# Delayed postglacial colonization of *Betula* in Iceland and the circum North Atlantic

David J Harning[1]*, Samuel Sacco[2], Kesara Anamthawat-Jónsson[3], Nicolò Ardenghi[1], Thor Thordarson[4], Jonathan H Raberg[1], Julio Sepúlveda[1,5], Áslaug Geirsdóttir[4], Beth Shapiro[2], Gifford H Miller[1,5]

[1]Institute of Arctic and Alpine Research, University of Colorado Boulder, Boulder, United States; [2]Department of Ecology and Evolutionary Biology, University of California Santa Cruz, Santa Cruz, United States; [3]Institute of Life and Environmental Sciences, University of Iceland, Reykjavik, Iceland; [4]Faculty of Earth Sciences, University of Iceland, Reykjavik, Iceland; [5]Department of Geological Sciences, University of Colorado Boulder, Boulder, United States

*For correspondence:
david.harning@colorado.edu

Competing interest: The authors declare that no competing interests exist.

**Abstract** As the Arctic continues to warm, woody shrubs are expected to expand northward. This process, known as 'shrubification,' has important implications for regional biodiversity, food web structure, and high-latitude temperature amplification. While the future rate of shrubification remains poorly constrained, past records of plant immigration to newly deglaciated landscapes in the Arctic may serve as useful analogs. We provide one new postglacial Holocene sedimentary ancient DNA (*seda*DNA) record of vascular plants from Iceland and place a second Iceland postglacial *seda*DNA record on an improved geochronology; both show Salicaceae present shortly after deglaciation, whereas Betulaceae first appears more than 1000 y later. We find a similar pattern of delayed Betulaceae colonization in eight previously published postglacial *seda*DNA records from across the glaciated circum North Atlantic. In nearly all cases, we find that Salicaceae colonizes earlier than Betulaceae and that Betulaceae colonization is increasingly delayed for locations farther from glacial-age woody plant refugia. These trends in Salicaceae and Betulaceae colonization are consistent with the plant families' environmental tolerances, species diversity, reproductive strategies, seed sizes, and soil preferences. As these reconstructions capture the efficiency of postglacial vascular plant migration during a past period of high-latitude warming, a similarly slow response of some woody shrubs to current warming in glaciated regions, and possibly non-glaciated tundra, may delay Arctic shrubification and future changes in the structure of tundra ecosystems and temperature amplification.

## eLife assessment

This **valuable** work on the paleovegetation history of Iceland has implications for the field of paleoecology, and the deglaciation history of Iceland and additional localities in Northern America and Europe via woody shrub colonization. The study uses a sedimentary ancient DNA metabarcoding approach to study this historic process. The strength of evidence is **solid**, with the methods (analysis of sedimentary DNA) and data analyses broadly supporting the claims.

## Introduction

As the planet warms, the Arctic is warming at 2–4 times the global rate (***Rantanen et al., 2022***). This warming is impacting Arctic ecosystems through a process known as 'shrubification,' or the northward expansion and increased height and density of deciduous woody vegetation (***Tape et al., 2006***;

*Myers-Smith et al., 2011*; *Elmendorf et al., 2012*; *Sweet et al., 2015*). Arctic shrubification alters the structure of tundra ecosystems, its regional biodiversity, food web structure, and nutrient availability (*Elmendorf et al., 2012*; *Fauchald et al., 2017*; *Collins et al., 2018*). Arctic shrubification also has implications for the climate system by reducing surface albedo (*Sturm et al., 2005*) and increasing atmospheric water vapor (*Pearson et al., 2013*) – both positive feedbacks that amplify high-latitude warming (*Thompson et al., 2022*). However, large uncertainties remain regarding the rate of future shrubification, such that the magnitude of these positive feedbacks is poorly constrained in predictive earth system models. One way to reduce future climate uncertainties is through the analysis of well-constrained paleoenvironmental records dating to warmer-than-present periods in Earth's recent history (e.g., *Tierney et al., 2020*).

Traditionally, macrofossils and pollen have formed the backbone of Quaternary paleovegetation reconstructions (*Birks, 2019*). However, macrofossils are not consistently preserved in sedimentary records, and pollen production is restricted to seed-bearing taxa, which can be obscured by long-distant transport from sources thousands of kilometers away (*Hyvärinen, 1970*; *Birks, 2003*; *Crump et al., 2019*). For example, in Iceland, exotic tree pollen from taxa never reported as macrofossils since the Miocene and Pliocene (i.e., *Pinus, Quercus, Corylus, Alnus,* and *Ulmus, Denk et al., 2011a*; *Denk et al., 2011b*) are found in Early Holocene sedimentary records following retreat of the Icelandic Ice Sheet (*Caseldine et al., 2006*; *Eddudóttir et al., 2015*). Hence, the first appearance of pollen need not reflect local colonization of a given taxon. Recent analytical advances in sedimentary ancient DNA (*seda*DNA) allow for more reliable vegetation reconstructions (e.g., *Capo et al., 2021*), due in part to their application in continuous lake sediment records that capture colonization times following local deglaciation. Taxa identified by DNA in lake surface sediment closely match those in catchment vegetation cover, confirming the local origin of DNA in lake sediment and improved reliability for taxa presence compared to pollen (*Sjögren et al., 2017*; *Alsos et al., 2018*). Past records of vascular plant *seda*DNA can therefore provide new insight into postglacial colonization patterns (*Crump et al., 2019*; *Alsos et al., 2022*) as well as the response of vegetation to changes in temperature during prior warm periods (*Clarke et al., 2020*; *Crump et al., 2021*; *Huang et al., 2021*).

In this study, we present one new Icelandic record (Stóra Viðarvatn) and update the age model of a second previously published record from Iceland (*Alsos et al., 2021*). For these two lakes, we also compare existing pollen and *seda*DNA datasets to highlight the ability of *seda*DNA in pinpointing the timing of each taxon's first appearance. We then compare *seda*DNA records from eight other published North Atlantic lakes that track local vegetation assemblages since deglaciation, with particular attention to postglacial colonization patterns of two woody plant families, Salicaceae and Betulaceae. Collectively, the North Atlantic vascular plant DNA datasets demonstrate that Salicaceae colonized deglaciated lands quickly, whereas Betulaceae was delayed by up to several thousand years. As these colonization patterns took place during a period of high-latitude warming, our datasets suggest that ongoing warming may also lead to different northward expansion rates of woody

**Table 1.** Lake site information.

| Site # | Lake name | Region | Latitude (°) | Longitude (°) | Reference |
|--------|-----------|--------|--------------|---------------|-----------|
| 1 | Lake Qaupat | Baffin Island | 63.68 | –68.20 | *Crump et al., 2019* |
| 2 | Bliss Lake | Greenland | 83.52 | –28.35 | *Epp et al., 2015* |
| 3 | Torfdalsvatn | Iceland | 66.06 | –20.38 | *Alsos et al., 2021* |
| 4 | Stóra Viðarvatn | Iceland | 66.24 | –15.84 | This study |
| 5 | Jodavannet | Svalbard | 77.34 | 16.02 | *Voldstad et al., 2020* |
| 6 | Lake Skartjørna | Svalbard | 77.96 | 13.82 | *Alsos et al., 2016b* |
| 7 | Langfjordvannet | Norway | 70.15 | 20.54 | *Alsos et al., 2022* |
| 8 | Eaštorjávri South | Norway | 70.43 | 27.33 | *Alsos et al., 2022* |
| 9 | Nordvivatnet | Norway | 70.13 | 29.01 | *Alsos et al., 2022* |
| 10 | Lake Ljøgottjern | Norway | 60.15 | 11.14 | *ter Schure et al., 2021* |

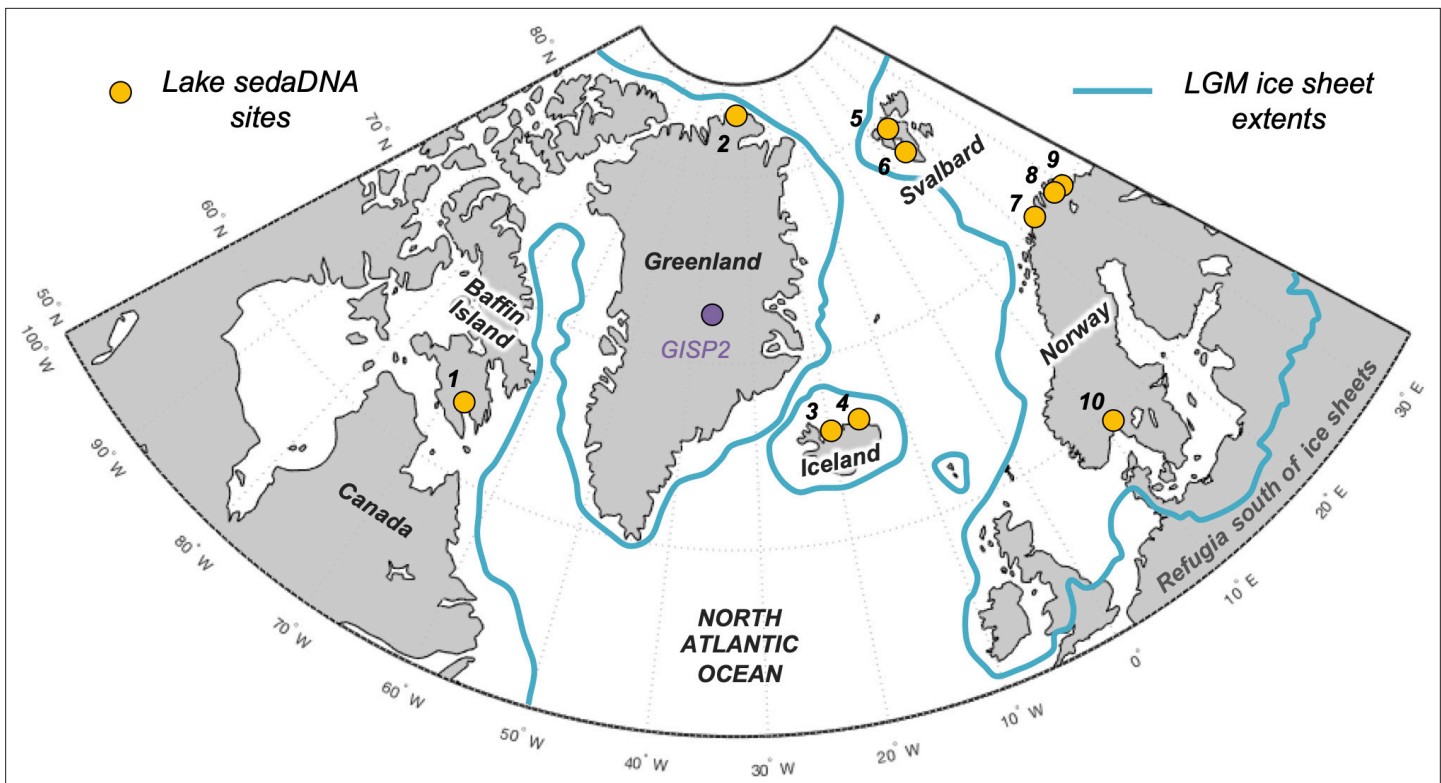

**Figure 1.** Overview map of the North Atlantic. Lake sedimentary ancient DNA (*sed*aDNA) sites with vascular plant histories since local deglaciation shown in yellow (see *Table 1* for site information). The extent of the Last Glacial Maximum (LGM) ice sheets is delineated in blue (*Batchelor et al., 2019*).

shrubs in glaciated regions, and possibly tundra regions as well. As a result, the rate of Arctic shrubification may be delayed, with important consequences for high-latitude ecosystems and temperature amplification.

## Results

### Stóra Viðarvatn, Iceland

Stóra Viðarvatn (#4, *Table 1*, *Figure 1*) is a large lake (surface area: 2.4 km², depth: 48 m) in northeast Iceland, with no preexisting records of Holocene vegetation change. In winter 2020, we recovered sediment core 20SVID-02 from 17.4 m water depth near the center of the lake. The core contains 893 cm of sediment, collected in 1.5 m increments. The basal 20 cm are laminated, clay-rich sediment, transitioning to organic gyttja by 873 cm depth, which characterizes the remainder of the record, interbedded with tephra (volcanic ash) deposits. The chronology of the Early to Middle Holocene portion of the record is based on five tephra layers (see 'Materials and methods'), of which the lowest, at 886.5 cm depth, is identified as the Askja S tephra (10,830 ± 57 BP, *Bronk Ramsey et al., 2015*), and the highest, at 334 cm depth, as Hekla 4 (4200 ± 42 BP, *Dugmore et al., 1995*). Because the sedimentation rate between the six tephra layers is essentially linear, we extrapolate that rate from the Askja S tephra layer 7 cm to the base of the sediment core at 893.5 cm depth, which provides a minimum deglaciation age of ~10,950 BP (*Figure 2A*).

Samples for *sed*aDNA analysis were taken from 75 equi-spaced depths throughout core 20SVID-02. Of these, 73 yielded amplifiable vascular plant DNA using the *trnL* P6 loop primer set. The two samples that failed (810 and 856.5 cm depth) were sampled within thick tephra layers with low organic content. Following data filtering (see 'Materials and methods'), the *trnL* dataset yields 15,738,794 total assigned reads, with an average of 209,851 reads per sample. qPCR cycle threshold ($C_T$) values, which reflect PCR efficiency and the quantity of suitable target sequences for amplification, average 32.1 across samples. While there is inter-sample variation in DNA reads and $C_T$ values, linear regressions

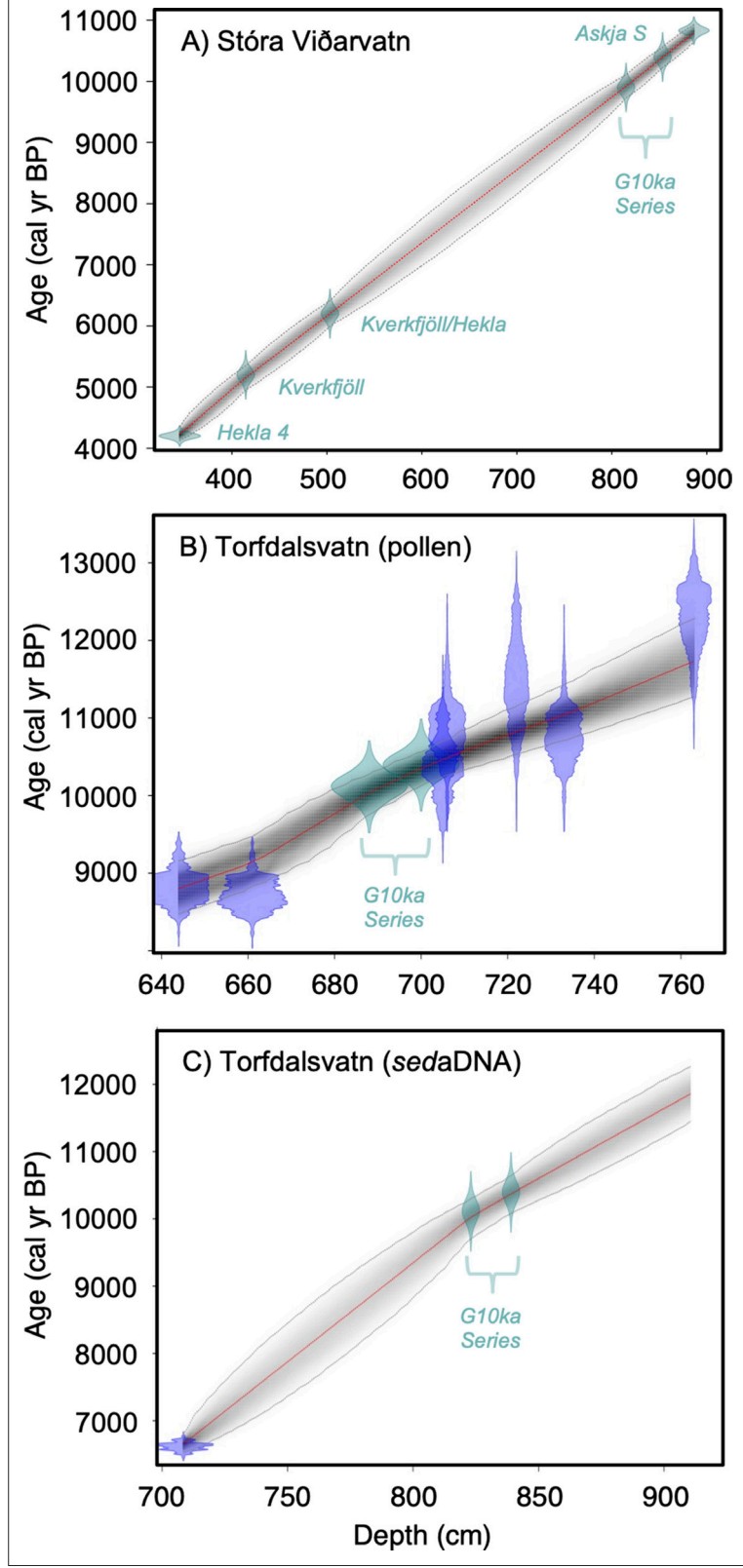

**Figure 2.** Lake sediment age models used in this study. (**A**) Stóra Viðarvatn's age model (this study), (**B**) Torfdalsvatn's pollen and macrofossil age model modified from *Rundgren, 1995*; *Rundgren, 1998* (see *Geirsdóttir et al., 2020*), and (**C**) Torfdalsvatn's sedimentary ancient DNA (*seda*DNA) age model modified from

*Figure 2 continued on next page*

*Figure 2 continued*

*Alsos et al., 2021*. Green points reflect tephra layers and blue points reflect calibrated $^{14}$C ages using IntCal20 (*Reimer et al., 2020*).

The online version of this article includes the following source data for figure 2:

**Source data 1.** Major oxide composition of tephra layers (mean and standard deviation) used for age control in the Stóra Viðarvatn lake sediment record.

show nearly stable temporal trends (*Figure 3*), which indicates that the availability of DNA to PCR amplification is consistent throughout the record. Metabarcoding technical quality (MTQ) and analytical quality (MAQ) scores are below suggested thresholds for low quality (0.75 and 0.1, respectively, *Rijal et al., 2021*) only in some samples during the Early Holocene (*Figure 3*). However, given that the scores correlate with species richness due to the latter values all falling below 30 (*Figure 3*), the low MTQ and MAQ scores are likely an artifact of the requirement that the 10 best represented barcode sequences are required for calculation (*Rijal et al., 2021*), and not an indication of low-quality DNA. As PCR amplification results in sequence read abundances that may not reflect original relative abundances in a sample (*Nichols et al., 2018*), we focus our discussion on taxa presence/absence, with a greater number of replicates (five maximum) representing a higher likelihood of taxa presence. We identified 53 taxa throughout Stóra Viðarvatn's lake sediment record (*Figure 4—source data 1*). Of the key woody shrub taxa, Salicaceae is present at the base of the core (892.5 cm depth, 10,850 BP) and is present nearly continuously through the core. Betulaceae first appears ~2300 y later (701.5 cm depth, 8560 BP), and has a sporadic presence until the Middle Holocene at ~5500 BP (*Figure 4A*).

## Torfdalsvatn, Iceland

Torfdalsvatn (#3, *Table 1*, *Figure 1*) is a small lake (surface area: 0.41 km$^2$, depth: 5.8 m) in north Iceland from which sediment cores have been analyzed since the early 1990s to create paleoecological records from pollen and macrofossils (*Björck et al., 1992*; *Rundgren, 1995*; *Rundgren, 1998*), and *sed*aDNA (*Alsos et al., 2021*). The paleovegetation records derived from pollen/macrofossils and *sed*aDNA were constructed from separate sediment cores that have independent age models derived from radiocarbon and tephra layers of known age. To temporally compare the two sediment records, radiocarbon ages need to be calibrated using the same calibration curve, correlative tephra layer ages must agree, and age model construction requires a common method (i.e., linear interpolation vs. Bayesian). For the pollen/macrofossil records (*Rundgren, 1995*; *Rundgren, 1998*), this required calibrating the radiocarbon ages using IntCal20 (*Reimer et al., 2020*) and application of a Bayesian age modeling approach that informs sediment rate changes with prior information (*Blaauw and Christen, 2011*; *Figure 2B*, *Geirsdóttir et al., 2020*). For the *sed*aDNA record (*Alsos et al., 2021*), we revised the tephra ages previously used to establish a more reliable Bayesian age model (see 'Materials and methods'). In doing so, the *sed*aDNA and pollen records feature similar linear sedimentation rates and are now synchronized by the G10ka Series tephra layers (*Figure 2C*). The revised age model for the Early Holocene portion of the pollen/macrofossil record (*Figure 2B*, *Geirsdóttir et al., 2020*) sets the timing of deglaciation, inferred from the onset of organic sedimentation at 801 cm depth, to 11,800 BP.

Differences on the order of centuries between the pollen/macrofossil and *sed*aDNA records may be artifacts of chronological uncertainty. This is because although the revised age models for the pollen/macrofossil and *sed*aDNA records feature common tephra layers, variable levels of proxy data resolution between the two records increase uncertainty when comparing the timing of inferred changes. Pollen of both Salicaceae (*Salix* spp.) and Betulaceae (*Betula* spp.) first appear by ~11,800 BP (*Figure 5*). The oldest *B. nana* macrofossil dates to ~9300 BP, similar to macrofossils of *S. herbacea* (~9400 BP) and *S. phylicifolia* (~9300 BP) (*Figure 5*). The *trnL* primed *sed*aDNA data indicate that Salicaceae did not colonize Torfdalsvatn's catchment until ~10,300 BP, whereas Betulaceae arrived later at ~9500 BP (*Figure 5*).

## Resolving Icelandic records of woody taxa colonization

Based on the original age model, Torfdalsvatn pollen records were interpreted to capture ecological changes associated with the abrupt climate oscillations between the Bølling-Allerød (14,700–12,900 BP)

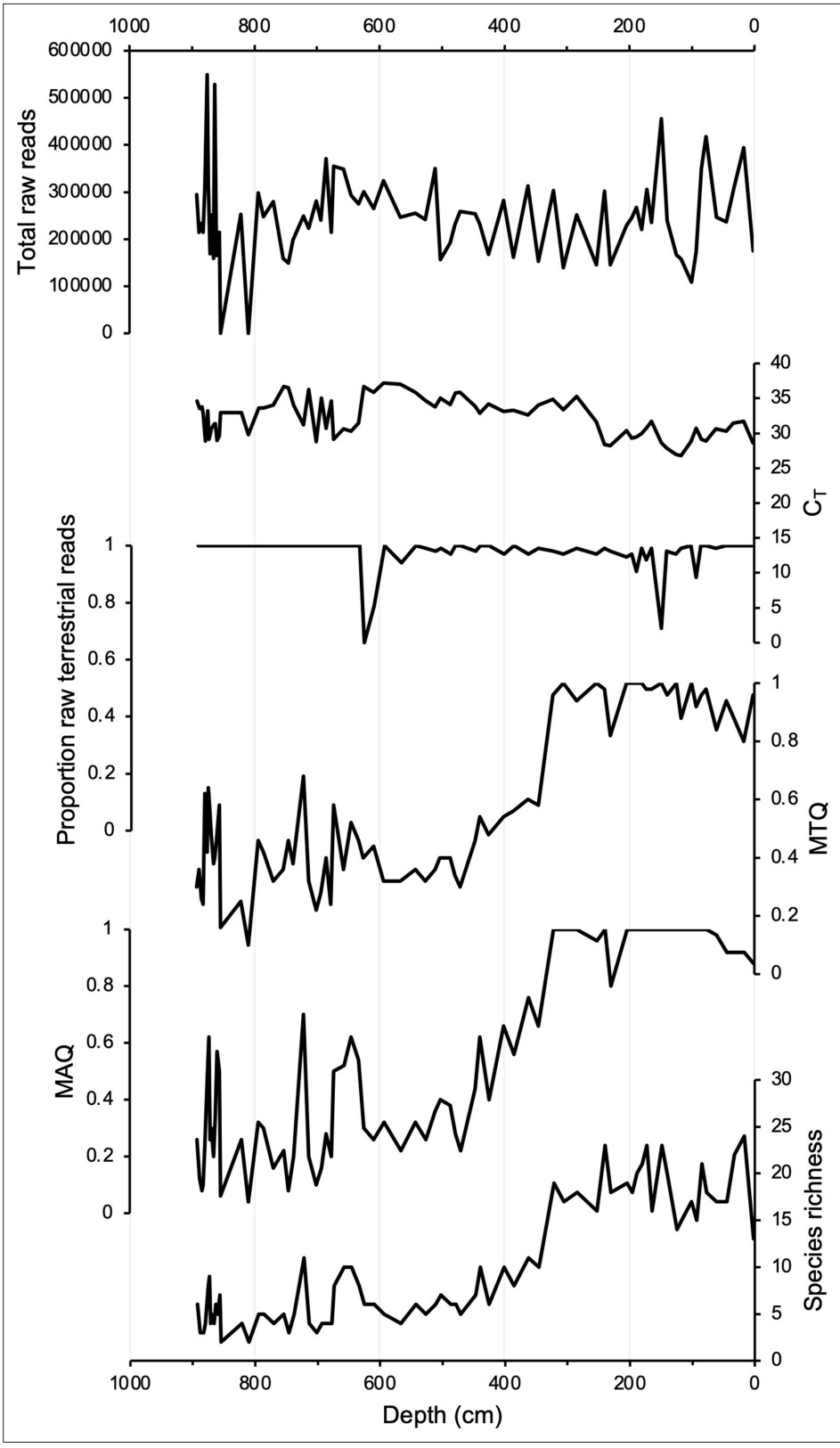

**Figure 3.** DNA quality assessment for Stóra Viðarvatn's entire Holocene record plotted against depth (cm). Total raw DNA reads, $C_T$ values, proportion raw terrestrial reads, metabarcoding technical quality (MTQ), metabarcoding analytical quality (MAQ), and species richness.

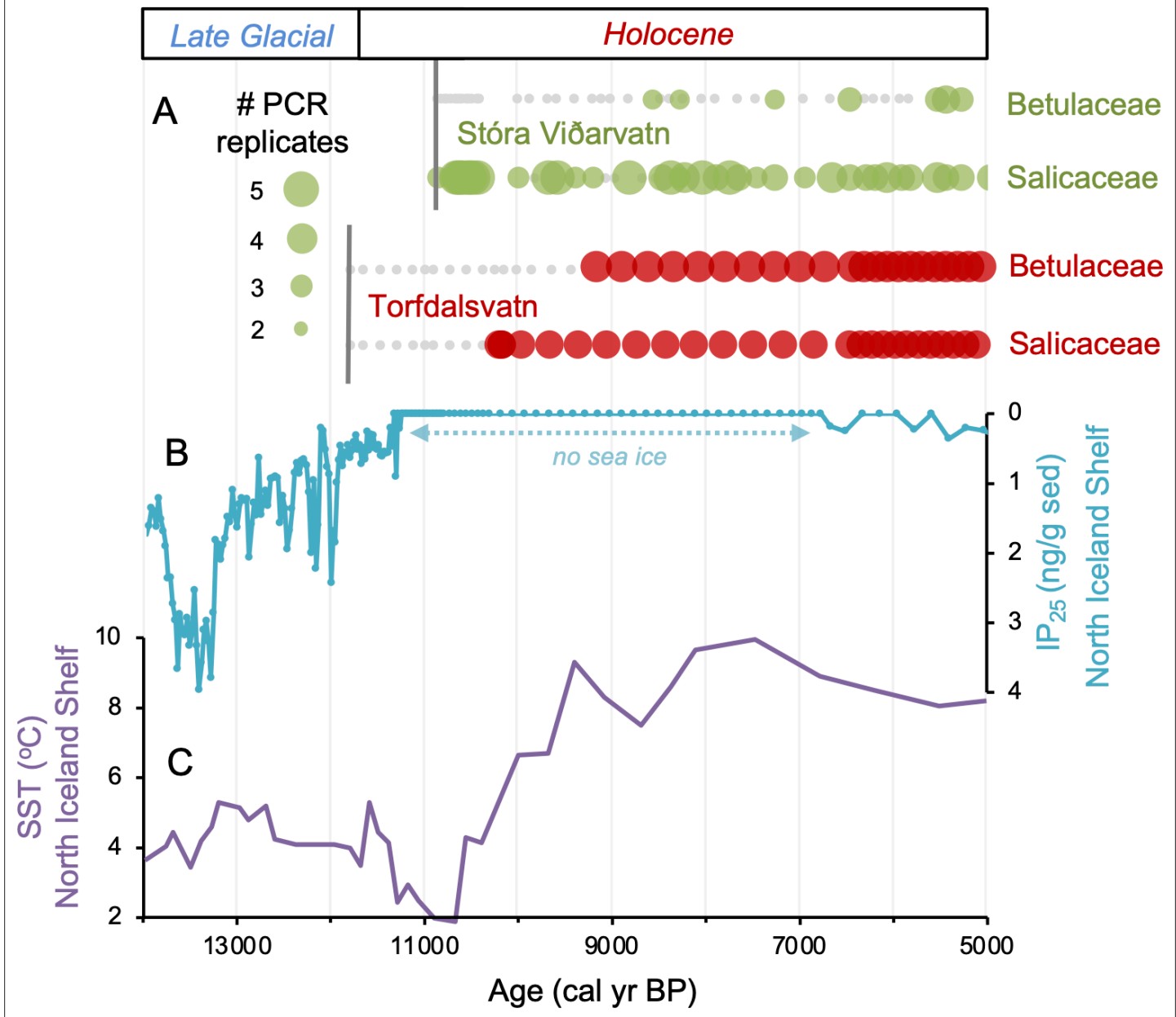

**Figure 4.** Comparison of paleoclimate and lake sedimentary ancient DNA (*sed*aDNA) records from Iceland. (**A**) Presence/absence *sed*aDNA record from Stóra Viðarvatn (green) and Torfdalsvatn (red) for Betulaceae and Salicaceae, where light gray markers denote that no taxa was detected. For Stóra Viðarvatn *sed*aDNA, the bubble size is proportional to the number of PCR replicates, and vertical dark gray lines denote the timing of each lake's deglaciation. (**B**) Biomarker-proxy record of sea ice from the North Iceland Shelf (inverted), where greater concentrations of IP$_{25}$ correspond to more sea ice (*Xiao et al., 2017*). (**C**) Diatom-based SST record from the North Iceland Shelf (*Sha et al., 2022*).

The online version of this article includes the following source data for figure 4:

**Source data 1.** Processed sedimentary ancient DNA (*sed*aDNA) data (number of replicates and reads) for all taxa identified in the Stóra Viðarvatn lake sediment record.

and Younger Dryas (12,900–11,700 BP) periods (*Björck et al., 1992*; *Rundgren, 1995*). However, recent calibration of the radiocarbon ages in the Torfdalsvatn cores from *Björck et al., 1992* and *Rundgren, 1995* suggests that the organic-rich portion of the record began less than 11,800 y ago (*Figure 2B*, *Geirsdóttir et al., 2020*). While there is sediment at deeper levels, it is described as silty clay (*Björck et al., 1992*) with sand and gravel (*Rundgren, 1995*). These characteristics suggest that the sediment likely originates from a rapidly deglaciating environment, perhaps with lingering ice sheet meltwater

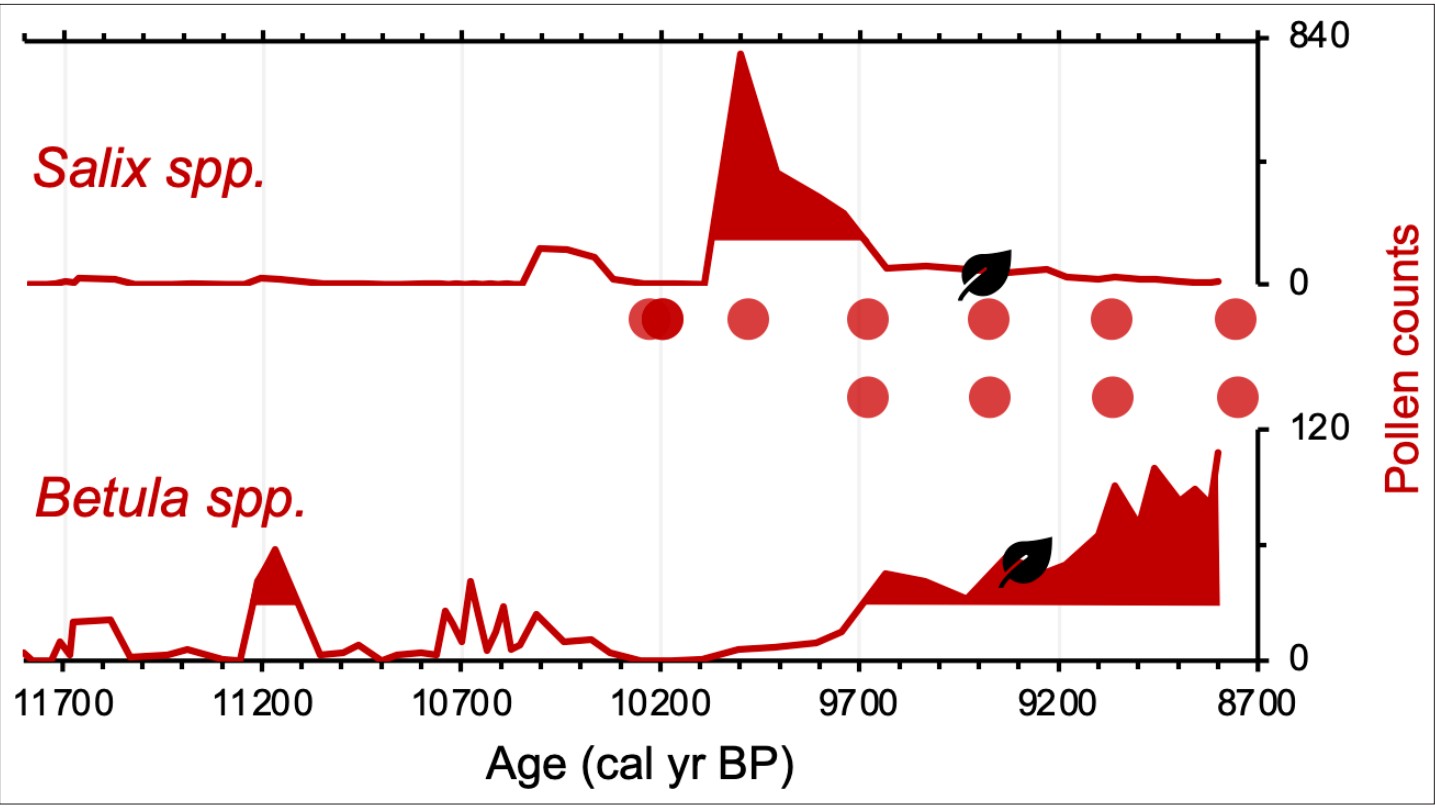

**Figure 5.** Simplified paleovegetation records from Torfdalsvatn for two taxa: *Salix* spp. and *Betula* spp. Shown are pollen counts (bold red lines), where shaded regions indicate values above the mean (***Rundgren, 1995***), first occurrence of taxa macrofossils (black leaves, ***Rundgren, 1998***), and DNA presence (red bubbles, ***Alsos et al., 2021***).

in the catchment and a rapid sediment accumulation rate that is difficult to constrain without secure age control. Therefore, we suggest that the maximum age for the final retreat of the Icelandic Ice Sheet from Torfdalsvatn's catchment, when organic-rich sedimentation began, is ~11,800 BP, several thousand years younger than previously proposed (***Björck et al., 1992***; ***Rundgren, 1995***). We use the updated age models to reconstruct the history of plant colonization in Iceland.

Differences in taxa presence/absence between the *seda*DNA and pollen records from Torfdalsvatn likely reflect different mechanisms of transport for DNA and pollen. *Seda*DNA for both Salicaceae and Betulaceae from Torfdalsvatn shows a colonization delay relative to deglaciation, whereas pollen from both taxa is present, albeit in low abundance, in the oldest sediment samples (***Figure 5***). This is most likely explained by the local catchment origin of *seda*DNA (***Sjögren et al., 2017***) versus pollen, which can be transported by wind from thousands of kilometers away (***Caseldine et al., 2006***; ***Eddudóttir et al., 2015***). Therefore, the occurrence of low pollen counts prior to the corresponding identification with *seda*DNA (***Alsos et al., 2021***) likely reflects long-distance transport from sources outside of Iceland rather than local establishment in the catchment – an inference supported by a lack of *Salix* and *Betula* macrofossils during this interval (***Figure 5***, ***Rundgren, 1998***). While poor DNA preservation could explain the delay in *seda*DNA, other vascular plants are identified before Salicaceae and Betulaceae DNA appear, suggesting adequate DNA preservation for PCR amplification (***Alsos et al., 2021***). Moreover, the corresponding sediment is rich in clay (***Rundgren, 1995***), which preferentially binds with and stabilizes DNA (***Kanbar et al., 2020***), making poor preservation unlikely. Therefore, we conclude that the colonization of Salicaceae and Betulaceae were delayed by 1500 and 2300 y, respectively, after local deglaciation of Torfdalsvatn's catchment.

Relative to Torfdalsvatn, the catchment of Stóra Viðarvatn deglaciated ~1000 y later at ~10,850 BP (***Figure 4A***). Based on our new *seda*DNA record, Salicaceae colonized the catchment rapidly, as it was identified in the lowermost sample at ~10,850 BP, whereas Betulaceae does not appear in the record until 8560 BP, more than 2000 y later. Considering $C_T$ values are relatively stable throughout the entire

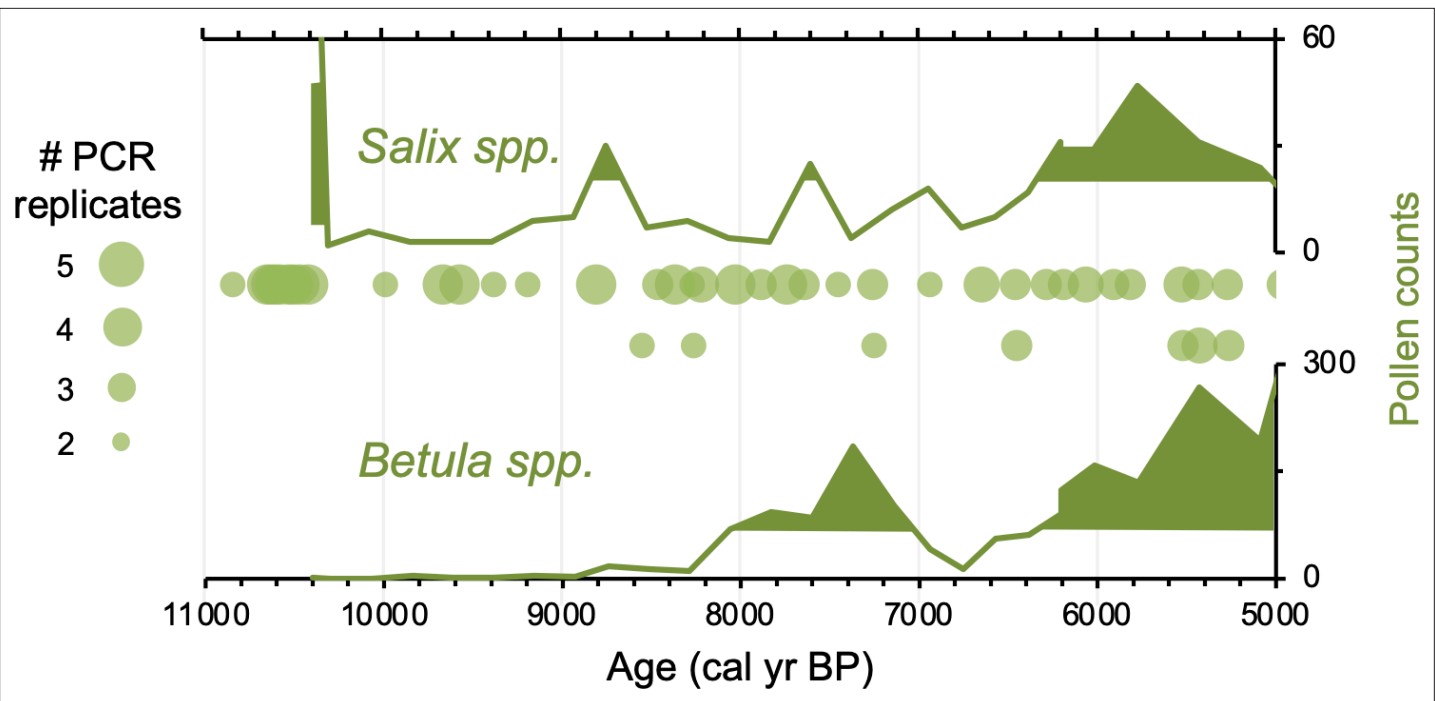

**Figure 6.** Simplified paleovegetation records from northeast Iceland for two taxa: *Salix* spp. and *Betula* spp. Shown are Ytra-Áland pollen counts (bold green lines), where shaded regions indicate values above the mean (*Karlsdóttir et al., 2014*) and DNA presence from Stóra Viðarvatn (green bubbles, this study), where the bubble size is proportional to the number of PCR replicates.

Holocene record (*Figure 3*), the delayed appearance of Betulaceae does not likely result from preservation bias. While Stóra Viðarvatn does not have any independent paleovegetation records, a pollen record from a nearby peat section Ytra-Áland in Thistilfjörður (*Karlsdóttir et al., 2014*), ~12 km to the east, provides a useful cross-examination of woody shrub colonization patterns. In the Ytra-Áland record, *Salix* pollen is found in high counts well above the mean in the lowermost sample dated to 10,400 BP, suggesting bona fide early colonization (*Figure 6*). For *Betula*, there is limited to no pollen present until ~8000 BP, after which there is sporadic presence as we observe with *sed*aDNA (*Figure 6*). The *Betula* pollen-limited period at Ytra-Áland is followed by the first peak of *Betula* presence in the total land pollen during ~7700–6800 BP. According to size differentiation between the two *Betula* species in Iceland (*Karlsdóttir et al., 2007*), the pollen recovered in this peak period is mostly from the dwarf birch *B. nana*, but with a fair amount of pollen from the shrub-like *B. pubescens*, suggesting arctic-alpine heath with stands of downy birch in the region (*Karlsdóttir et al., 2014*). Despite the coarser resolution of Ytra-Áland's age model, which only has two tephra layer dates bounding the timing of the first appearance of these taxa (10,400 BP and 7050 BP, *Karlsdóttir et al., 2014*), there is reasonable agreement between lake and peat records for the timing of arrival and subsequent Early Holocene establishment of Salicaceae and Betulaceae in northeast Iceland.

Based on the *sed*aDNA records from Torfdalsvatn and Stóra Viðarvatn, we find that as the Icelandic Ice Sheet receded and exposed new land Salicaceae colonized the landscape. This occurred 0–1700 y after inferred deglaciation, whereas Betulaceae colonization was further delayed by a few thousand years (~2300 y in both cases). As Torfdalsvatn is the oldest dated terrestrial site of deglaciation in Iceland, this implies that ~1700 y was required for Salicaceae to establish a viable population on the island. By the time Stóra Viðarvatn deglaciated, it is likely that other portions of north Iceland had also deglaciated allowing Salicaceae to more efficiently colonize new regions of Iceland. The similar timing of Betulaceae colonization relative to deglaciation in the two catchments (~2300 y) but different times of deglaciation (11,800 and 10,850 BP) suggests that the evolution of catchment conditions for Torfdalsvatn and Stóra Viðarvatn may be additional factors controlling the local colonization of this taxon on Iceland. The earlier presence of low concentrations of Salicaceae and Betulaceae pollen in Torfdalsvatn is most likely derived from long-distance wind dispersal from outside Iceland.

## Discussion

### Postglacial *seda*DNA records from the circum North Atlantic

Eight additional records of vascular plant *seda*DNA from lake sediments located around the North Atlantic that were glaciated during the Last Glacial Maximum (LGM) are available to compare with the Icelandic data (see 'Materials and methods' for details). These include records from Baffin Island (n = 1, *Crump et al., 2019*), northern Greenland (n = 1, *Epp et al., 2015*), Svalbard (n = 2, *Alsos et al., 2016b*; *Voldstad et al., 2020*), northern Norway (n = 3, *Rijal et al., 2021*; *Alsos et al., 2022*), and southern Norway (n = 1, *ter Schure et al., 2021*; *Figure 1*). All eight records have age control derived from calibrated radiocarbon ages. Except for Baffin Island, we infer minimum ages of deglaciation based on the age of the oldest sediment.

### Baffin Island

Lake Qaupat (#1, *Figure 1*) is a small lake (surface area: 0.08 km², depth: 9.2 m) situated at 35 m asl on southern Baffin Island. Cosmogenic ¹⁰Be exposure dating of Lake Qaupat's impounding moraine constrains the timing of deglaciation to 9100 ± 700 BP (*Crump et al., 2019*). Lake Qaupat and most of its catchment were below sea level until 7700 ± 300 BP when postglacial isostatic recovery raised the basin above the ocean and its catchment was available for vascular plant colonization. Salicaceae is present in the record shortly after isolation from the ocean at 7400 BP, but no vascular plants are documented by *seda*DNA in the older marine sediment. The first appearance of Betulaceae *seda*DNA occurred at 5900 BP, more than 1000 y after Salicaceae and 2000 y after the lake and its catchment were above sea level (*Crump et al., 2019*).

### Greenland

Bliss Lake (#2, *Figure 1*) is a small lake (depth: 9.8 m) situated at 17 m asl on the northern coastline of Greenland (Peary Land). Bliss Lake initially deglaciated 11,000 BP (*Olsen et al., 2012*) and the first appearance of Salicaceae *seda*DNA is found in low read counts (<100) in sediment dated to 10,800 BP but only in higher read counts beginning at 7400 BP (*Epp et al., 2015*). As the lake was inundated by marine water between 10,480 and 7220 BP (*Olsen et al., 2012*), 7400 BP is interpreted to be the most reliable date for Salicaceae colonization in the catchment as sea level regressed and isolated the lake from the ocean (*Epp et al., 2015*).

### Svalbard

Jodavannet (#5, *Figure 1*) is a small lake (depth: 6.4 m) situated at 140 m asl on the east coast of Wijdefjorden on northern Spitsbergen. Regional cosmogenic exposure dating suggests the area began to deglaciate between 14,600 and 13,800 BP (*Hormes et al., 2013*), although the base of Jodavannet's sediment record suggests a minimum timing of lake deglaciation by 11,900 BP (*Voldstad et al., 2020*). Salicaceae *seda*DNA was first found at 9800 BP in Jodavannet, whereas Betulaceae was not detected with *seda*DNA metabarcoding (*Voldstad et al., 2020*). Lake Skartjørna (#6, *Figure 1*) is a small lake (surface area: 0.10 km², depth: 7.5 m) situated at 65 m asl on the west coast of Spitsbergen. As sea level fell due to glacial isostatic adjustment, Lake Skartjørna likely became isolated from the sea at ~13,000 BP (*Landvik et al., 1987*), although the base of the recovered sediment record, which provides a minimum deglaciation age, is dated to 8600 BP (*Alsos et al., 2016b*). Salicaceae *seda*DNA is identified in Lake Skartjørna's basal sample at 8600 BP and Betulaceae is first identified at 7000 BP (*Alsos et al., 2016b*).

### Norway

Langfjordvannet (#7, *Figure 1*) is a small lake (surface area: 0.55 km², depth: 34.8 m) situated at 66 m asl on the coast of northern Norway (*Rijal et al., 2021*). Langfjordvannet deglaciated by 16,150 BP, Salicaceae is first identified in *seda*DNA by 15,500 BP, and Betulaceae by 10,000 BP (*Alsos et al., 2022*). Eaštorjávri South (#8, *Figure 1*) is a small lake (surface area: 0.06 km², depth: 5.4 m) situated at 260 m asl near the coast of northern Norway (*Rijal et al., 2021*). Eaštorjávri South deglaciated by 11,060 BP, with Salicaceae *seda*DNA identified in the basal sample at 11,060 BP and consistently present above that level. Betulaceae is identified, albeit with low read counts, in the basal sample as well, but not identified in the overlying sample. Since Betulaceae *seda*DNA is not present with

substantial read counts until 9950 BP (**Alsos et al., 2022**), we conservatively set this as Betulaceae's first appearance in Eaštorjávri South. Nordvivatnet (#9, **Figure 1**) is a small lake (surface area: 0.05 km², depth: 13.3 m) situated at 82 m asl on the coastline of northern Norway (**Rijal et al., 2021**). Nordvivatnet deglaciated by 12,880 BP, with Salicaceae *seda*DNA identified in the basal sample. Betulaceae *seda*DNA is identified in the basal sample as well but below our cutoff for taxa presence (see 'Materials and methods'). The first sample with high read counts indicative of Betulaceae presence is at 12,080 BP (**Alsos et al., 2022**), which we conservatively set as the timing of the taxa's first appearance in Nordvivatnet. Finally, Lake Ljøgottjern (#10, **Figure 1**) is a small lake (surface area: 0.02 km², depth: 18 m) situated at 185 m asl in southeastern Norway. Lake Ljøgottjern deglaciated by 9300 BP and both Salicaceae and Betulaceae *seda*DNA were first detected by 7900 BP (**ter Schure et al., 2021**).

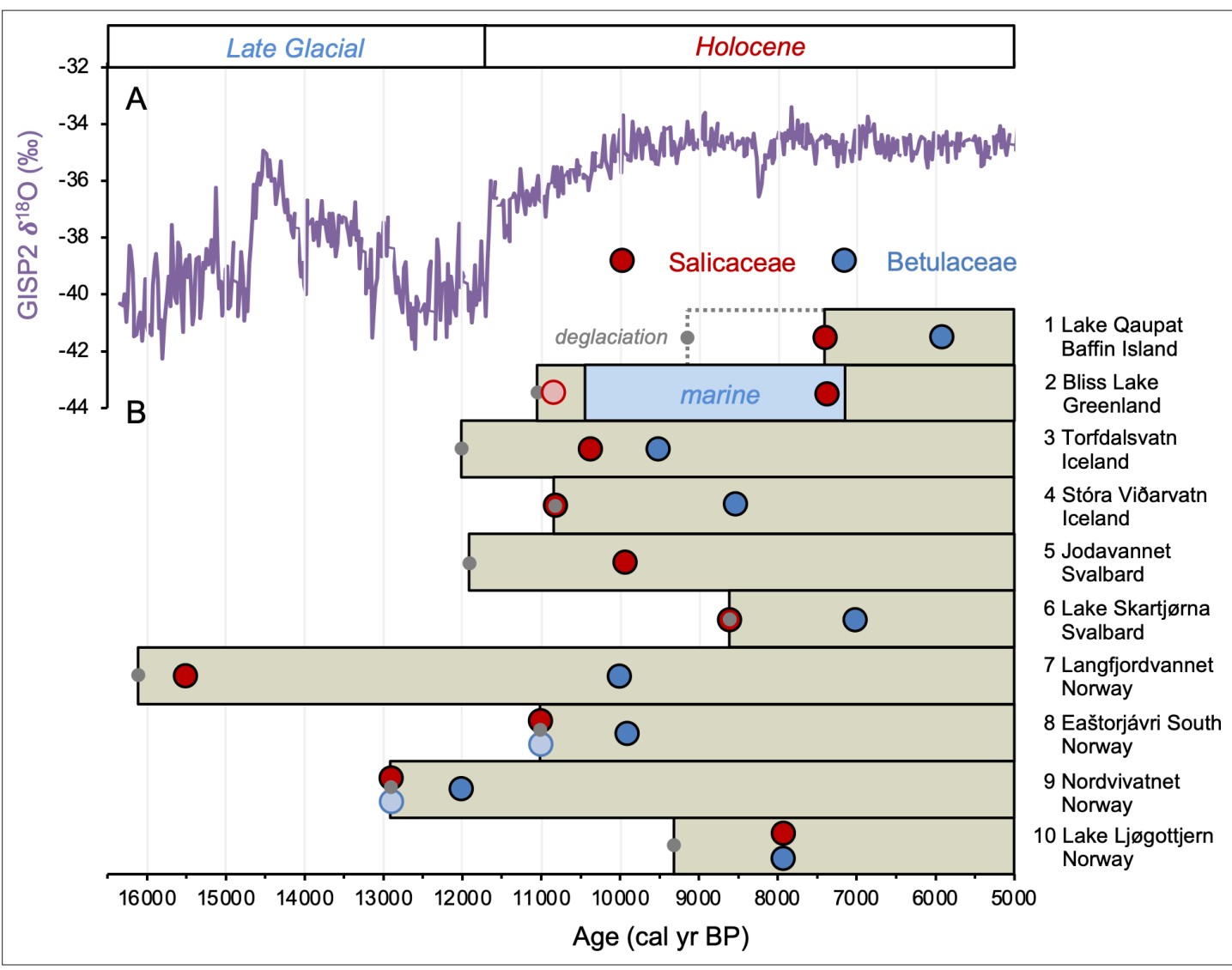

**Figure 7.** Timing of postglacial Salicaceae and Betulaceae colonization in the circum North Atlantic at 10 locations (see **Figure 1** and **Table 1**). (**A**) GISP2 δ¹⁸O values reflective of regional North Atlantic temperature variability (**Seierstad et al., 2014**) and (**B**) lake *seda*DNA records of vascular plants, where the brown boxes reflect the extent of sedimentary record and the red/blue circles denote the first appearance of Salicaceae/Betulaceae (**Epp et al., 2015**; **Crump et al., 2019**; **Alsos et al., 2016b, Alsos et al., 2021**; **Alsos et al., 2022**; **Voldstad et al., 2020**; **Rijal et al., 2021**; **ter Schure et al., 2021**; this study). Light red and blue circles denote the presence of a low number of taxa reads, which we conservatively do not interpret to reflect genuine presence (e.g., **Epp et al., 2015**). Gray circles denote the minimum timing of local deglaciation inferred from basal sediment core ages, and in the case of Baffin Island, cosmogenic radionuclide dating of the lake's impounding moraine (**Crump et al., 2019**).

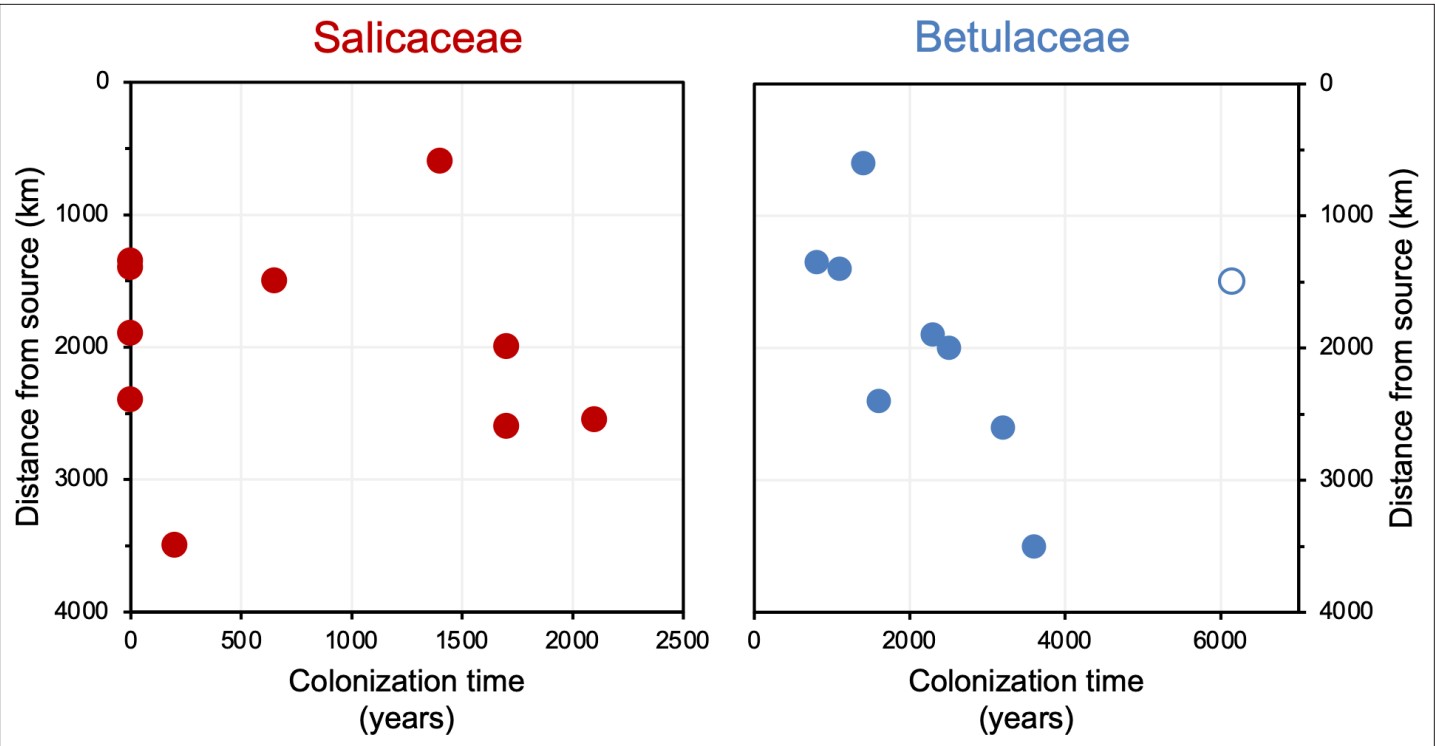

**Figure 8.** Colonization time after deglaciation (years) versus lake distance from the closest possible source south of the Last Glacial Maximum (LGM) ice sheet margin (km). For all circum North Atlantic sites, Salicaceae is shown in red and Betulaceae in blue. Data point for Langfjordvannet is open for Betulaceae.

### Betulaceae colonization is delayed relative to Salicaceae in the circum North Atlantic

The timing of postglacial Salicaceae and Betulaceae colonization varies across the North Atlantic regions, with first appearance dates ranging from 15,500 BP in Norway to 5900 BP on Baffin Island (*Figure 7*). Salicaceae appears immediately after inferred deglaciation at 4 out of 10 sites, while colonization dates for the other 6 locations range from 200 to 2100 y after deglaciation and exhibit no clear spatio-temporal pattern (*Figure 8*). For Betulaceae, colonization times range from 800 to 6150 y after deglaciation, with sites closest to source refugia south of the ice sheets (*Figure 1*) having generally shorter colonization delays (*Figure 8*), consistent with seedling viability having a higher probability of success with shorter distance from source (*Nathan, 2006*). The one exception is Langfjordvannet, in northern Norway (site #7, *Figure 1*, *Alsos et al., 2022*), which deglaciated at 16,150 BP (*Figure 7*). Considering that Betulaceae only arrives during the Holocene Epoch (last 11,700 y) across all North Atlantic sites, the anomalously long delay of Betulaceae to Langfjordvannet suggests that this taxon may have required the Holocene's relatively stable climate, as reflected by Greenland oxygen isotope records (*Figure 7*, *Seierstad et al., 2014*), to successfully establish itself. Alternatively, the colonization delay of Betulaceae to Langfjordvannet may reflect the slow development of stable soils, which are required for Betulaceae establishment, due to the lake's steep catchment (*Otterå, 2012*). In contrast to Betulaceae, Salicaceae appears to have been an efficient colonizer to Langfjordvannet during the colder and less stable climate of the Late Glacial period and earliest Holocene (*Figure 7*, *Seierstad et al., 2014*). Finally, except for Lake Ljøgottjern in southern Norway (site #10, *Figure 1*) where Betulaceae and Salicaceae apparently colonize at about the same time (*ter Schure et al., 2021*), all other sites show that Betulaceae colonization is delayed by 800–5500 y relative to Salicaceae colonization dates (*Figure 7*).

Several possibilities may contribute to the delay in Betulaceae colonization compared to Salicaceae colonization. First, the northernmost Salicaceae species reach bioclimate subzone A (1–3°C July temperature), whereas the northernmost Betulaceae species rarely go beyond bioclimate subzone D (8–9°C July temperature) (*Walker et al., 2005*). The different environmental tolerances may explain

why Salicaceae generally arrives earlier when regional temperatures are still climbing compared to the timing of Betulaceae colonization when warmth is more broadly established (*Figure 7*). In support, *Alsos et al., 2022* showed that at least for Fennoscandian *seda*DNA vascular plant records, temperature was one of the dominant controls on postglacial dispersal patterns. Second, species diversity is greater for the Salicaceae family compared to Betulaceae family in the North Atlantic, and conventional DNA metabarcoding techniques cannot identify *seda*DNA to the species level. Iceland, for example, has four native Salicaceae taxa, that is, three dwarf, cold-tolerant arctic species (*S. herbacea, S. lanata,* and *S. arctica*) and one shrub-like species (*S. phylicifolia*), and only two Betulaceae (*B. nana* and *B. pubescens*) (*Kristinsson, 2008*). The suite of Salicaceae taxa can collectively tolerate a wider range of environments than Betulaceae, including disturbed, wet, and nutrient-limited substrate characteristic of deglacial environments (e.g., *S. herbacea*; *Beerling, 1998*). Third, the modes of reproduction for Salicaceae and Betulaceae are different. Salicaceae is clonal, meaning that plants mostly reproduce asexually by vegetative spreading, which is advantageous when colonizing new landscapes (e.g., *Beerling, 1998*). Betulaceae propagates instead by seed and therefore requires a larger number of founder individuals for successful establishment (*Atkinson, 1992*; *de Groot et al., 2007*). Moreover, Betulaceae seeds are generally heavier than Salicaceae seeds (dry mass, *B. nana* = 0.12–0.30 mg, *de Groot et al., 2007*; *S. herbacea* = 0.07–0.14 mg, *Beerling, 1998*). This mass difference would provide a higher probability of long-distance wind dispersal for Salicaceae compared to Betulaceae and establishment of the pioneer plants.

Finally, modern observations broadly replicate the records of delayed woody taxa colonization in the North Atlantic. Studies tracking the migration of Salicaceae and Betulaceae in the forefields of retreating glaciers and to new islands also show a rapid and faster colonization of Salicaceae compared to Betulaceae (*Whittaker, 1993*; *Magnússon et al., 2009*; *Burga et al., 2010*; *Synan et al., 2021*). As an example, following the formation of the volcanic island Surtsey in 1964 CE, Salicaceae (*S. herbaceae, S. lanata,* and *S. phylicifolia*) colonized between 1995 and 1999 CE and Betulaceae has yet to arrive (*Magnússon et al., 2009*). Complimentary studies tracking soil and nutrient development at proglacial sites in the North Atlantic (*Vilmundardóttir et al., 2014*) suggest that Salicaceae is environmentally unconstrained (*Whittaker, 1993*; *Glausen and Tanner, 2019*), whereas Betulaceae requires more developed soil for its ultimate viability (*Synan et al., 2021*). Collectively, these studies support the environmental constraints known for each woody taxon and reinforce the hypothesis that Salicaceae is a more efficient pioneer than Betulaceae, particularly in newly deglaciated landscapes.

## Environmental dispersal mechanisms of Salicaceae and Betulaceae

Coupling the timing of postglacial woody shrub colonization in the North Atlantic with genetic approaches provides a valuable opportunity to identify probable environmental dispersal mechanisms (e.g., wind, sea ice, driftwood, and birds) for long-distance transport. Previously, *Alsos et al., 2015* compared amplified fragment length polymorphism (AFLP) data from living plants from Iceland, Greenland, Svalbard, the Faroe Islands, and Jan Mayen to AFLP data from the same species from potential source regions. Genetic distances suggest that *Betula* (sp. *nana* and *pubescens*) and *Salix herbacea* populations in most locations descend from ancestors in a Western European LGM refugia (*Alsos et al., 2009*; *Eidesen et al., 2015*), and that each taxa underwent a broad-fronted and rapid migration to newly deglaciated regions (*Alsos et al., 2015*). *B. nana* on Svalbard likely descends from a northwest Russian source (*Alsos et al., 2007*, *Alsos et al., 2015*), and populations on Baffin Island and western Greenland originated from North America (*Alsos et al., 2009*; *Eidesen et al., 2015*). Molecular analysis of Icelandic *Betula* based on chloroplast haplotypes indicates that *Betula* may have postglacially immigrated to Iceland from northern Scandinavia (essentially *B. nana*) and northwestern Europe (*B. pubescens*) (*Thórsson et al., 2010*). However, the relative roles of wind, sea ice, driftwood, and birds as transport vectors remain an open question.

Sea ice has been suggested to be a dispersal mechanism for postglacial vascular plants in the Arctic (*Alsos et al., 2016a*). While seasonal sea ice persisted throughout the Holocene around Svalbard (*Pieńkowski et al., 2021*) and northern Greenland (*Syring et al., 2020*), general ocean circulation patterns require source populations of vascular plants in Arctic Siberia, the dominant source region for sea ice export to the North Atlantic, for sea ice to be a viable vector. Considering that *B. nana* on Svalbard likely descends from a Russian refugia (*Alsos et al., 2007*, *Alsos et al., 2015*), sea ice is a reasonable transport mechanism for this location. However, genetic data from Salicaceae

(*S. herbacea*) on Baffin Island, Greenland, and Svalbard suggest that the modern populations from these locations descend from source populations in either mid-latitude North America (south of the Laurentide Ice Sheet) or Western Europe (*Alsos et al., 2007*, *Alsos et al., 2009*), and neither region exported sea ice in the Early Holocene. This leaves wind and birds as likely mechanisms for *S. herbacea* transport to Baffin Island, Greenland, and Svalbard during the Holocene. North of Iceland, proxy evidence shows that sea ice disappeared by ~11,200 BP (*Figure 4B*, *Xiao et al., 2017*) when surface ocean currents were established bringing warm Atlantic waters to the North Iceland Shelf (*Figure 4C*, *Eiriksson et al., 2000*; *Sha et al., 2022*), nearly 500 y before the colonization of the first woody taxon on the island (*Figure 4A*). Thus, with the absence of sea ice around Iceland's coastline during woody plant colonization, wind and/or birds remain the likeliest dispersal mechanisms for Iceland, similar to other Arctic regions.

## Future outlook

As the planet continues to warm, glacier and ice sheet mass loss is accelerating (*Hugonnet et al., 2021*), providing new territory for woody shrubs to colonize – the Greenland Ice Sheet currently occupies 1.7 million $km^2$ (*Morlighem et al., 2017*) with >400,000 $km^2$ more covered by glaciers elsewhere in the Arctic (*RGI Consortium, 2017*). Global warming will also allow woody plants to migrate poleward across unglaciated Arctic tundra (e.g., *Myers-Smith et al., 2011*), which occupies a substantially larger area compared to glaciated regions (5 billion $km^2$, *Walker et al., 2005*). In this study, we demonstrate that Salicaceae can migrate into deglaciating North Atlantic habitats rapidly, whereas Betulaceae may be delayed by millennia. However, the ability of these taxa to successfully follow the rapid pace of ongoing range shifts in non-glaciated tundra (e.g., Alaska and Siberia) remains an open question (e.g., *Davis and Shaw, 2001*; *Thuiller et al., 2008*). This is partially due to different baseline conditions of the tundra ecosystem compared to glacial regions upon deglacial warming, which already contain established ecosystems and nutrient-rich soils bound in the permafrost (e.g., *Heijmans et al., 2022*).

While the causes and rates of present-day climate change differ from the preceding postglacial period (e.g., *Tierney et al., 2020*), the factors that we identify as potentially driving the colonization patterns (i.e., environmental tolerance, modes of reproduction, and soil development) are relevant in both scenarios. As a result, circum North Atlantic shrubification in deglaciating regions may be prolonged – ultimately dampening the rate of change, at least temporarily, in regional biodiversity and food web structure, as well as high-latitude temperature amplification. An important next step for constraining future shrubification rates is developing detailed paleoecological reconstructions of plant dynamics in multiple tundra regions (e.g., *Clarke et al., 2020*; *Huang et al., 2021*), continuing these efforts in glaciated regions, and pairing plant histories with independent local records of climate, such as temperature and precipitation. The latter paleoclimate reconstructions will enable a more detailed assessment of plant colonization lags relative to climate (e.g., *Crump et al., 2019*). By pairing paleoecological and paleoenvironmental records from formerly glaciated and tundra regions and integrating these with predictive models (e.g., *Braconnot et al., 2012*), the community will be able to reduce uncertainties and better quantify future shrubification rates, which are essential to inform ecological predictions for policymakers.

## Materials and methods
### Lake sediment cores and age control

Stóra Viðarvatn (66.24°N, 15.84°W) is a large (2.4 $km^2$), deep lake (48 m) located at 151 m asl in northeast Iceland (*Figure 1*, *Axford et al., 2007*). In winter 2020, we recovered a composite 8.93-m-long sediment core (20SVID-02) from 17.4 m water depth using lake ice as a coring platform. The sediment was collected in seven drives of ~150 cm each. The core sections were subsequently stored at 4°C before opening for sediment subsampling. The chronology of 20SVID-02 is based on five marker tephra layers (volcanic ash) of known age: Askja S, G10ka Series, Kverkfjöll/Hekla, Kverkfjöll, and Hekla 4 tephra layers, with ages of 10,830 ± 57 BP (*Bronk Ramsey et al., 2015*), 10,400–9900 BP (*Óladóttir et al., 2020*), 6200 BP (*Óladóttir et al., 2011*), 5200 BP (*Óladóttir et al., 2011*), and 4200 ± 42 BP (*Dugmore et al., 1995*), respectively. Each tephra layer was sampled along the vertical axis, sieved to isolate glass fragments between 125 and 500 µm, and embedded in epoxy plugs. Individual glass

shards were analyzed at the University of Iceland on a JEOL JXA-8230 election microprobe using an acceleration voltage of 15 kV, beam current of 10 nA, and beam diameter of 10 µm. The international A99 standard was used to monitor for instrumental drift and maintain consistency between measurements. Tephra origin was then assessed following the systematic procedures outlined in *Jennings et al., 2014* and *Harning et al., 2018*. Briefly, based on $SiO_2$ wt% vs. total alkali ($Na_2O + K_2O$) wt%, we determine whether the tephra volcanic source is mafic (tholeiitic or alkalic), intermediate, and/or rhyolitic. From here, we objectively discriminate the source volcanic system through a detailed series of bielemental plots produced from available compositional data on Icelandic tephra (*Harning et al., 2018*). The major oxide composition of each tephra layer is provided in *Figure 2—source data 1*. A Bayesian age model was generated using the R package rbacon and default settings (*Blaauw and Christen, 2011*; *Figure 2A*).

Torfdalsvatn (66.06°N, 20.38°W) is a small (0.4 km²), shallow (5.8 m) lake located at 52 m asl in north Iceland (*Figure 1*, *Rundgren, 1998*). Paleovegetation records from this lake are based on pollen, macrofossil assemblages (*Björck et al., 1992*; *Rundgren, 1995*; *Rundgren, 1998*; *Rundgren and Ingólfsson, 1999*), and *seda*DNA (*Alsos et al., 2021*). Sediment cores from Torfdalsvatn were also recovered by a joint Colorado-Iceland team in 2004, 2008, 2012, and 2020. In modifying the lake sediment core chronologies, we use the recently calibrated age model (*Figure 2B*, *Geirsdóttir et al., 2020*) for the pollen/macrofossil records (*Björck et al., 1992*; *Rundgren, 1995*) that uses the IntCal20 calibration curve (*Reimer et al., 2020*). We note that the ~12,000 BP Vedde Ash has been previously identified in this record based on major oxide composition (*Björck et al., 1992*). However, the age of the Vedde Ash is stratigraphically too old for the radiocarbon-based chronology, suggesting that this tephra layer likely represents ash from a separate and subsequent volcanic eruption that produced the Vedde Ash, of which several possible correlations have been identified in Iceland and mainland Europe (e.g., *Pilcher et al., 2005*; *Kristjánsdóttir et al., 2007*; *Matthews et al., 2011*; *Geirsdóttir et al., 2022*). For the *seda*DNA record, the Early Holocene portion of lake sediment core (*Alsos et al., 2021*) features two age control points, the so-called 'Saksunarvatn Ash' (10,267 ± 89 BP) and a graminoid radiocarbon age (6620 ± 120 BP, *Alsos et al., 2021*). To better compare with the pollen/macrofossil record, we generated a new age model for the *seda*DNA record (*Alsos et al., 2021*) using the lowermost radiocarbon age and the upper and lower limits of the main ~20-cm-thick G10ka Series/Saksunarvatn tephra unit (10,400–9900 BP) following the recent recognition of multiple associated tephra layers and dating of these layers in Iceland (*Harning et al., 2018*; *Harning et al., 2019*; *Óladóttir et al., 2020*). The new Bayesian age model was generated using the R package rbacon and default settings (*Blaauw and Christen, 2011*) with the IntCal20 calibration curve (*Reimer et al., 2020*; *Figure 2C*).

## DNA metabarcoding

We subsampled the Stóra Viðarvatn sediment core halves immediately upon splitting in a dedicated clean lab facility with no PCR products in the Trace Metal Lab, University of Colorado Boulder. All surfaces and tools were treated with 10% bleach and 70% ethanol before use. Personnel wearing complete PPE took a total of 75 samples for *seda*DNA analysis using clean metal spatulas. Samples were then stored in sterile Falcon tubes at 4°C until DNA extraction and metabarcoding. Sampling resolution was based upon a preliminary age model with a sample estimated every 150 y.

We performed DNA extraction and metabarcoding library generation in a dedicated ancient DNA laboratory at the Paleogenomics Lab, University of California Santa Cruz. While working in this laboratory, we followed commonly used practices for minimizing contamination of ancient and degraded samples described previously (*Fulton and Shapiro, 2019*). We initially cleaned all instruments and surfaces using 10% bleach and 70% ethanol before UV irradiation for at least 1 hr. From each homogenized sample of the core, we aliquoted two 50 mg subsamples and isolated DNA following the column purification method described in *Rohland et al., 2018* using Binding Buffer D and eluting 50 µl volumes. We pooled extraction eluates derived from the same original core sample prior to subsequent experiments and included one extraction control (containing no sample) for every 12 samples processed to assess the presence of external contaminants or cross-contamination.

We performed quantitative PCR (qPCR) to determine the ideal dilution of extract to add to the metabarcoding PCR, as well as the ideal number of PCR cycles. We added 2 µl of neat or diluted extract to wells containing 1× QIAGEN Multiplex master mix, as well as a final concentration of 0.16 µm of

**Table 2.** Primer sequences used in metabarcoding and qPCR experiments.

| Primer | Sequence 5′–3′ |
| --- | --- |
| truseq_trnL_g | ACACTCTTTCCCTACACGACGCTCTTCCGATCTGGGCAATCCTGAGCCAA |
| truseq_trnL_h | GTGACTGGAGTTCAGACGTGTGCTCTTCGATCT TTGAGTCTCTGCACCTATC |

the forward and reverse trnL primer and 0.12× SYBR Green I Dye. Cycling conditions were as follows: a 15 min 95°C activation, followed by 40 cycles of 94°C for 30 s, 57°C for 30 s, and 72°C for 60 s. We tested 1:10 and 1:100 dilutions of extract to water alongside neat extract. We used cycle threshold ($C_T$) values to compare PCR efficiency between dilutions and determined the ideal cycle number for the metabarcoding PCR as the number of cycles at which the PCR curve reached the plateau phase.

We performed metabarcoding PCR on each extract using the best dilution as determined by qPCR and versions of the chloroplast *trnL* primer set (*Taberlet et al., 2007*) that contained attached truseq adapters (*Table 2*). We used five replicate PCRs for each primer set and for each sample, with reaction conditions as follows: 2 µl of diluted or undiluted extract in 1× QIAGEN Multiplex Mix and a 0.16 µM concentration of each primer at a total volume of 25 µl. Thermocycling began with 15 min at 95°C, followed by a number of cycles (as determined by qPCR) at 94°C for 30 s, 50–57°C for 90 s, 72°C for 60 s. We used one negative control PCR (no sample) alongside every 16 samples to assess the possibility of contamination during the PCR set up. Extraction controls also underwent metabarcoding. We cleaned reactions using SPRI select beads and eluted them into a final volume of 20 µl.

We then used indexing PCR to attach the full truseq adapter and unique identifier. We added 5 µl of cleaned PCR product from the metabarcoding PCR to 1× Kapa Hifi master mix and dual-unique truseq style indices at a concentration of 0.4 µM. Thermocycling began with an activation step at 98°C for 3 min, followed by 8 cycles of 98°C for 20 s, 65°C for 30 s, 72°C for 40 s, and a final extension at 72°C for 2 min. We cleaned reactions using SPRI select beads and quantified DNA concentration using the NanoDrop 8000 Spectrophotometer. We pooled PCR replicates from all samples in equimass ratios and sequenced the pools on an Illumina Nextseq 550 platform. Sequencing was performed using Illuminas v3 chemistry and a 150-cycle mid-output sequencing kit with paired end reads of 76 base pair length.

We processed raw fastq files and performed taxonomic assignment using the Anacapa Toolkit (*Curd et al., 2019*) and the ArcBorBryo library, which contains 2280 sequences of Arctic and boreal vascular plants and bryophytes (*Sønstebø et al., 2010*). Briefly, we removed adapter sequences from the raw fastq data using cutadapt and trimmed low-quality bases using FastX-toolkit before reads were separated into paired and unpaired files (*Hannon, 2010*; *Martin, 2011*). We then used dada2 for further filtering of chimeric sequences and for merging of paired reads (*Callahan et al., 2016*.). We dereplicated identical sequences present in the data using dada2 and generated ASV (amplicon sequence variant) tables and fastq files for each of the read types (paired merged, paired unmerged, unpaired forward, and unpaired reverse). ASVs were assigned taxonomy by globally aligning to a CRUX database generated from the ArcBorBryo library using Bowtie2 before assignment with Anacapa's custom BLCA algorithm (*Langmead and Salzberg, 2012*). Outputted taxonomy site frequency tables containing taxonomic assignments and read counts were subsequently analyzed using R. ASV sequences for Salicaceae and Betulaceae identified in our samples are included in *Table 3*.

To minimize the chance of false positives, we retained only sequences that (1) were not detected in negative controls, (2) had 100% match to sequences in the library, (3) had a minimum of 10 reads per PCR replicate and occurred in a minimum of two of five PCR replicates, and (4) had a minimum of 100 reads across all PCR replicates (*Table 4*). We verified non-native taxa identified by this pipeline via comparison with the NCBI nucleotide database using BLAST (*Sayers et al., 2022*) (http://www.

**Table 3.** ASV (amplicon sequence variant) sequences identified in our samples for Salicaceae and Betulaceae.

| Taxon | ASV sequence |
| --- | --- |
| Salicaceae | ATCCTATTTTTCGAAAACAAACAAAGGTTCATAAAGACAGAATAAGAATACAAAAG |
| Betulaceae | ATCCTGTTTTCCGAAAACAAATAAAACAAATTTAAGGGGTTCATAAAGTGAGAATAAAAAAG |

**Table 4.** Total read data remaining after each data filtering step.

| Step | Total read data remaining |
|---|---|
| Raw | 18,653,132 |
| Minimum 10 reads per PCR replicate, occurred in 2 out of 5 PCR replicates, and minimum 100 reads across PCR replicates | 17,303,887 |
| Non-native taxa and blank contaminants | 15,909,615 |

ncbi.nlm.nih.gov/blast/) and removed these due to the low likelihood of correct taxonomic assignment. Finally, DNA quality scores were calculated following *Rijal et al., 2021*.

## Published *seda*DNA datasets

To place the Icelandic plant *seda*DNA datasets in the context of the circum North Atlantic, we compiled all existing lake *seda*DNA records of vascular plants from this region that were formally glaciated during the LGM (*Figure 1*, *Table 1*). These include records from Baffin Island (n = 1, *Crump et al., 2019*), northern Greenland (n = 1, *Epp et al., 2015*), Svalbard (n = 2, *Voldstad et al., 2020 Alsos et al., 2016b*; *Voldstad et al., 2020*), northern Norway (n = 3, *Rijal et al., 2021*; *Alsos et al., 2022*), and southern Norway (n = 1, *ter Schure et al., 2021*). To ensure that all records were comparable, we excluded seven records from northern Norway: Sandfjorddalen, Horntjernet, Gauptjern, Jøkelvatnet, Kuutsjärvi, Nesservatnet, Sierravannet (*Rijal et al., 2021*; *Alsos et al., 2022*). Nesservatnet and Sierravannet contained only Late Holocene records and thus do not capture local postglacial colonization patterns. A lack of supporting proxy records (e.g., carbon content) for Sandfjorddalen and Horntjernet meant it was unclear whether these two records capture the complete postglacial period. For Gauptjern, the inferred basal age (8776 BP) is inconsistent with the reconstructed timing of deglaciation (~11,000 BP, *Hughes et al., 2016*), which suggests the initial postglacial interval may be missing. Finally, both Jøkelvatnet and Kuutsjärvi were impacted by glacial meltwater during the Early Holocene when woody taxa were first identified (*Wittmeier et al., 2015*; *Bogren, 2019*), and thus the inferred timing of plant colonization is probably confounded in this unstable landscape by periodic pulses of terrestrial detritus. Unless otherwise specified, the timing of deglaciation at each site is inferred from the basal age of the sediment record.

## Acknowledgements

This study was supported by NSF ARCSS #1836981. We kindly thank Sveinbjörn Steinþórsson and Þór Blöndahl for lake coring assistance at Stóra Viðarvatn, Mats Rundgren for sharing Torfdalsvatn's pollen datasets, Thomas Marchitto for access to the Trace Metal Lab at the University of Colorado Boulder, and Martha Raynolds and Helga Bültmann for valuable discussion. Publication of this article was funded by the University of Colorado Boulder Libraries Open Access Fund. We thank two anonymous reviewers for their constructive comments that improved the final version of our manuscript.

## Additional information

### Funding

| Funder | Grant reference number | Author |
|---|---|---|
| National Science Foundation | 1836981 | Gifford H Miller, Áslaug Geirsdóttir, Julio Sepúlveda |

The funders had no role in study design, data collection and interpretation, or the decision to submit the work for publication.

### Author contributions

David J Harning, Data curation, Formal analysis, Investigation, Visualization, Methodology, Writing – original draft, Writing – review and editing; Samuel Sacco, Formal analysis, Investigation, Methodology, Writing – review and editing; Kesara Anamthawat-Jónsson, Nicolò Ardenghi, Investigation, Writing – review and editing; Thor Thordarson, Formal analysis, Investigation, Writing – review and editing; Jonathan H Raberg, Investigation; Julio Sepúlveda, Conceptualization, Funding acquisition,

Writing – review and editing; Áslaug Geirsdóttir, Beth Shapiro, Gifford H Miller, Conceptualization, Supervision, Funding acquisition, Writing – review and editing

### Author ORCIDs
David J Harning ⓘ https://orcid.org/0000-0002-2648-1346
Kesara Anamthawat-Jónsson ⓘ http://orcid.org/0000-0002-3133-2185
Nicolò Ardenghi ⓘ http://orcid.org/0000-0002-6305-0588
Áslaug Geirsdóttir ⓘ http://orcid.org/0000-0003-3125-0195
Beth Shapiro ⓘ http://orcid.org/0000-0002-2733-7776

Joint Public Review: https://doi.org/10.7554/eLife.87749.3.sa1
Author Response https://doi.org/10.7554/eLife.87749.3.sa2

---

## Additional files

### Supplementary files
• MDAR checklist

### Data availability
Data associated with this study is publicly available in the manuscript and *Figure 2—source data 1* and *Figure 4—source data 1*.

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
