## [Editor Report · eLife assessment]

This **valuable** work on the paleovegetation history of Iceland has implications for the field of paleoecology, and the deglaciation history of Iceland and additional localities in Northern America and Europe via woody shrub colonization. The study uses a sedimentary ancient DNA metabarcoding approach to study this historic process. The strength of evidence is **solid**, with the methods (analysis of sedimentary DNA) and data analyses broadly supporting the claims.

---

## [Referee Report · Joint Public Review]

The revised version of the manuscript "Delayed postglacial colonization of Betula in Iceland and the circum North Atlantic" by Harning et al. investigates the colonization of shrubs during the Late Pleistocene/Holocene in Northern America and Europe by comparing published sedimentary ancient DNA (sedaDNA) records (and pollen data) with a new sedaDNA record from Island. The manuscript aims to identify shrub colonization patterns, discusses their drivers and evaluates the importance of shrubification under future warming.

The revised version improved the clarity of methods and discussion and results presented are more convincing.

However, parts of the methods (e.g. assessment of blanks and data filtering) and results (e.g. visualization of plant community data) could still be polished, and the figures should be improved to increase the clarity of the manuscript.

---

## [Author Response]

The following is the authors’ response to the current reviews.

We thank Reviewer 1 for their time reviewing our revised manuscript and appreciate their thoughtful suggestions for further clarity. In regard to the public review statement, "However, parts of the methods (e.g. assessment of blanks and data filtering) and results (e.g. visualization ofplant community data) could still be polished, and the figures should be improved to increase the clarity of the manuscript", we have made small modifications in the text and figures during production of the Version of Record to address these important suggestions.

The following is the authors’ response to the original reviews.

**Reviewer #1 (Public Review):**
This manuscript compiles the colonization of shrubs during the Late Pleistocene in Northern America and Europe by comparing plant sedimentary ancient DNA (sedaDNA) records from different published lake sediment cores and also adds two new datasets from Island. The major findings of this work aim to illuminate the colonization patterns of woody shrubs (Salicaceae and Betulaceae) in these sediment archives to understand this process in the past and evaluate its importance under future deglaciation and warming of the Arctic.

We greatly appreciate the time and detailed consideration of our manuscript by Reviewer 1. Our responses to individual comments are highlighted in blue, with the original comments provided by the reviewer in black.

The strength of evidence is solid as methods (sedimentary DNA) and data analyses broadly support the claims because the authors use an established metabarcoding approach with PCR replicates (supporting the replicability of PCR and thereby proving the occurrence of Salicaeae and Betulaceae in the samples) and quantitative estimation of plant DNA with qPCR (which defines the number of cycles used for each PCR amplification to prevent overamplification). However, the extraction methods need more explanation and the bioinformatic pipeline is not well-known and needs also some further description in the main text (not only referring to other publications).

Thank you for bringing this to our attention. We have now provided greater detail on our extraction methods and bioinformatic pipeline.

The authors compare their own data with previously published data to indicate the different timing of shrubification in the selected sites and show that Salicaceae occurs always like a pioneer shrub after deglaciation, followed by Betaluaceae with a various time lag. The successive colonization of Salicaceae followed by Betulaceae is explained by its differences in environmental tolerance, the time lag of colonization in the compared records is e.g. explained by varying distance to source areas.However, there are some weaknesses in the strength of evidence because full sedaDNA plant DNA assessment, quality of the sedaDNA data (relative abundance and richness of sedaDNA plant composition) and results from Blank controls (for sedaDNA) are not fully provided. I think it is important to show how the plant metabarcoding in general worked out, because it is known that e.g. poor richness can be indicative of less preserved DNA and a full plant assessment (shown in the supplement) would be more comprehensive and would likely attract a larger readership.

Thank you for bringing these important points to our attention. The DNA dataset including the full taxa assemblage will be included with the manuscript upon publication and apologize for not including it during the review process. This dataset will also include information on positive and negative blanks used for quality control. Following suggestions from Reviewer 2, we have now also calculated some recently proposed DNA quality metrics (Rijal et al., 2021), which collectively support our earlier conclusions that our record is of sufficient quality to draw the current conclusions. We hope that the inclusion of the complete DNA dataset will indeed draw a larger readership.

Further, it would allow us to see the relative abundance in changes of plants and would make it easier to understand if the families Salicaeae and Betulaceae are a major component of the community signal. Further, the possibility to reach higher taxonomic resolution with sedaDNA compared to pollen or to facilitate a continuous record (which is different from macrofossils) is not discussed in the manuscript but should be added. Also, the taxonomic resolution within these families in the discussed datasets would be of interest, also on the sequence type level if tax. assignments are similar.

Thank you for these suggestions. We have focused on these two families as it is known from numerous pollen records and floras that they are the major component of the vascular plant communities in the regions investigated. Betula (birch) and Salix (willow) are indeed the most dominant woodland shrubs of the tundra biome, which covers expansive areas of the Arctic. For example, in Iceland natural woodlands, which cover 1.5% of the total land area, are composed of 80% birch shrubs (Snorrason et al. 2016, Náttúrufræðingurinn 86). Salix mixes in with Betula, especially around wet sites. Species from both genera are common and wide-spread throughout Iceland, but dwarf and cold tolerant species thrive best on the highland or at glacial sites, while shrub-like species are more common on the lowland, coastal area and in sheltered valleys. Flora of Iceland (http://www.floraislands.is/PDF-skjol/Checklist-vascular.pdf) lists Betula as the only genus of Betulaceae native to Iceland (page 79/80) and Salix as the major genus of Salicaceae (page 82-85), although Populus tremula (Salicaceae) exists in the wild but is rare (perhaps just a countable number of trees/shrubs in the whole country). The point is that, for Iceland, Betulaceae is Betula and Salicaceae is Salix, meaning that our sedaDNA method has the taxonomic resolution at the genus level. And with the help of pollen analysis of the site near Stóra Viðarvatn (the novel sedaDNA work of the present paper), i.e., Ytri-Áland site (Karlsdóttir et al. 2014), it is possible to interpret our results even to the species level, which we have only mention in the discussion. It has been suggested that matching sedaDNA results with botanical knowledge about the study site and the vegetation history (local reference database) is one way to increase taxonomic resolution of the sedaDNA approach (e.g. Elliott et al. 2023, Quaternary 6,7). In the same way we find our sedaDNA analysis having sufficient resolution to answer the questions asked in the present study. For the future, although we do not include it in the discussion this time, it should be possible to increase the taxonomic resolution of plant metabarcoding by priming multiple genes simultaneously like that is described as a proof of concept by Foster et al. (2021, Front Ecol Evol 9: 735744). In the revised version of the manuscript, we have now expanded on the power of sedaDNA in terms of increased taxonomic resolution and application in continuous lake sediment records in the introduction of the manuscript. Following Reviewer 2’s suggestion, we have now included the sequences used for taxonomic assignment in the supplement information.

Another important aspect is how the abundance/occurrence of Salicaceae is discussed. Many studies on sedaDNA confirm an overrepresentation of this family due to better preservation in the sediment, far-distance transport along rivers, or preferences of primers during amplification etc. As this family is the major objective of this study, such discussion should be added to the manuscript and data should be presented accordingly.

Thank you for raising this point. The reviewer is indeed correct that Salicaceae is typically overrepresented in read abundance compared to other vascular plant taxa in sedaDNA studies. However, as we mention in the Results and Interpretation section for Stóra Viðarvatn “As PCR amplification results in sequence read abundances that may not reflect original relative abundances in a sample (Nichols et al., 2018), we focus our discussion on taxa presence/absence,” we do not place weight on the indeed greater relative abundance of Salicaceae in our own dataset. As such, this different relative abundance of plant taxa reads should not influence the conclusions drawn in the manuscript.

I also miss more clarity about how the authors defined the source areas (refugia) of the shrubs. If these source areas are described in other literature I suggest to show them in a map or so. Further, it should be also discussed and explained more in detail which specific environmental preferences these families have, this is too short in the introduction and too unspecific. Also, it would be beneficial to show relative abundances rather than just highlighted areas in the Figures and it would allow us to see if Salicaeae will be replaced by Betulaceae after colonizing or if both families persist together, which might be important to understand future development of shrubs in these areas.

Thank you for allowing us to clarify. As the regions studied with the lake sediment records shown in this manuscript were all covered by extensive ice sheets during the Last Glacial Maximum (LGM, Fig. 1), plant refugia and source areas must have been located somewhere south of the ice sheet margins. Thus, we calculate our distance to source as the minimum distance from a lake site to land beyond the extent of the ice sheet during the LGM. This has now been clarified in the text and highlighted in Fig. 1. We have also added in the discussion molecular results from Thórsson et al. (2010, J Biogeogr 37) on possible source origins of Betula in Iceland. Details on taxa environmental preferences have now been expanded upon in the Discussion section where we explore the various trait-based factors that may influence the relative differences in colonization timing between Salicaceae and Betulaceae. We have now also edited Figs. 3 and 4 to include PCR replicates instead of highlighted bars to better compare the DNA and pollen datasets from Iceland.

The author started a discussion about shrubification in the future, but a more defined evaluation and discussion of how to use such paleo datasets to predict future shrubification and its consequences for the Arctic would give more significance to the work.

Thank you for this suggestion and allowing us to expand on potential future changes. We have now edited this final section of the paper to provide a little more detail on how we envision these records being used to predict future shrubification and climate change.

**Reviewer #1 (Recommendations For The Authors):**
I list some more specific details here.You speak about "read counts", I guess you used relative abundance of read counts, you should state it like this.

Thank you for allowing us to clarify. The data that we refer do in terms of read counts is from the previously published studies in the circum North Atlantic. The data provided from these studies is raw read counts, and not relative abundance.

Line 100: What do you mean here: "temperature changes in prior warm periods"?

Thank you for allowing us to clarify. We have rephrased to sentence to “higher temperature in prior warm periods”, which we hope is clearer for the reader.

Line 134: How is DNA diluted by minerogenic sediment? Did the sedimentation rate increase? Typically minerogenic input should be beneficial for DNA preservation.

Thank you for allowing us to clarify. These samples were primarily comprised of tephra glass with minimal organic content. While we agree that minerogenic sediment is generally beneficial for DNA preservation, the predominance of inorganics (tephra) that fell from the sky, rather than being washed into the lake from the landscape, would not carry organic sediment with it. We have rephrased the sentence to make this clearer.

I would suggest adding more citations to the text (for example statements in lines 106, 110, 368)

Thank you for the suggestion. The manuscript has been edited accordingly.

Better divide your discussion part: discussion about dispersal mechanisms occur in both sections. Maybe you could divide it into environmental factors for colonization and traitbased factors (only an idea).

Thank you for the suggestion. We have now edited the second dispersal section to “Environmental dispersal mechanisms” to be clearer about our focus on factors such as wind, sea ice, and birds that may transport the seeds across the North Atlantic. The previous section retains the trait-based factors that may influence relative timing in colonization between Salicaceae and Betulaceae.

Which type of sequencing did you use, paired-end 76bp is unknown to me.

Methods have now been edited to clarify this, along with details related to extraction methods as requested in the Public Review.

**Reviewer #2 (Public Review):**
Harding et al have analysed 75 sedaDNA samples from Store Vidarvatn in Iceland. They have also revised the age-depth model of earlier pollen, macrofossil, and sedaDNA studies from Torfdalsvatn (Iceland), and they review sedaDNA studies for first detection of Betulaceae and Salicaceae in Iceland and surrounding areas. Their Store Vidarvatn data are potentially very interesting, with 53 taxa detected in 73 of the samples, but only results on two taxa are presented. Their revised age-depth model cast new light on earlier studies from Torfdalsvatn, which allows a more precise comparison to the other studies. The main result from both sedaDNA and the review is that Salicaceae arrives before Betulaceae in Iceland and the surrounding area. This is a well-known fact from pollen, macrofossil, and sedaDNA studies (Fredskild 1991 Nordic J Bot, Birks & Birks QSR 2014, Alsos et al. 2009, 2016, 2022) and as expected as the northernmost Salix reach the Polar Desert zone (zone A, 1-3oC July temperature) whereas the northernmost Betula rarely goes beyond the Southern Tundra (zone D, 8-9 oC July temperature, Walker et al. 2005 J. Veg. Sci., Elven et al.2011 http://panarcticflora.org/ ).

We greatly appreciate the time and detailed consideration of our manuscript by Reviewer 2. Our responses to individual comments are highlighted in blue, with the original comments provided by the reviewer in black.

While we agree that previous studies have indeed indicated a relative delay in Betula colonization relative to Salix, most of these have relied on pollen and macrofossil evidence, which are complicated to use as proxies for the first appearance of a given taxa (see our Introduction in the main manuscript). A few studies have shown this also with sedaDNA (e.g., Alsos et al., 2022), which is a more robust proxy for a plant taxa’s presence, but these have been limited geographically (e.g., northern Fennoscandia). In our study, we show that this pattern is reflected in 10 different lakes across the North Atlantic, emphasizing the broad nature of Betula’s delayed colonization relative to other woody shrubs, such as Salix.

My major concern is their conclusion that lag in shrubification may be expected based on the observations that there is a time gap between deglaciation and the arrival of Salicaceae and between the arrival of Salicaceae and Betulaceae. A "lag" in biological terms is defined as the time from when a site becomes environmentally suitable for a species until the species establish at the site (Alexander et al. 2018 Glob. Change Biol.). The climate requirement for Salicaceae highly depends on species. In the three northernmost zones (A-C), it appears as a dwarf shrub, and it only appears as a shrub in the Southern Tundra (D) and Shrub Tundra (E) zone, and further south it is commonly trees. Thus, Salicaceae cannot be used to distinguish between the shrub tundra and more northern other zones, and therefore cannot be used as an indicator for arctic shrubification. Betulaceae, on the other hand, rarely reach zone C, and are common in zone D and further south. Thus, if we assume that the first Betulaceae to arrive in Iceland is Betula nana, this is a good indicator of the expansion of shrub tundra. Thus, if they could estimate when the climate became suitable for B. nana, they would have a good indicator of colonisation lags, which can provide some valuable information about time lags in shrub expansion (especially to islands). They could use either independent proxy or information from the other species recorded in sedaDNA to reconstruct minimum July temperature (see e.g. Parducci et al. 2012a+b Science, Alsos et al. 2020 QSR).

We appreciate the reviewer’s insight into the implications of our use of the word “lag”. Indeed, as we do not have site-specific climate timeseries for each lake record, we have adjusted our wording to “delay”, which we believe is more general and descriptive of our observations. We recognize the importance of independent paleotemperature records for each lake, but these are not yet available for all records, so we prefer to keep our study focused on the delay instead. In addition, we prefer not to derive temperature records from the vegetation sedaDNA records, as these are not independent and will incorporate changes driven by additional factors, such as soil and light (e.g., Alsos et al., 2022). We have added some text to the final section on Future Outlook that elaborates on the need for complimentary records of past climate to pair with paleoecological records of colonization. We hope that this motivates the community to pursue these lines of research that we agree are needed.

The study gives a nice summary of current knowledge and the new sedaDNA data generated are valuable for anyone interested in the post-glacial colonisation of Iceland. Unfortunately, neither raw nor final data are given. Providing the raw data would allow re-analysing with a more extensive reference library, and providing final data used in their publication will for sure interest many botanists and palaeoecologist, especially as 73 samples provide high time resolution compared to most other sedaDNA studies.

Finally, the raw and final data, including blank controls, used in our study for Stóra Viðarvatn will ultimately be provided with the manuscript’s publication. We apologize for not including it with the original submission.

**Reviewer #2 (Recommendations For The Authors):**
Line 112-113: Difference in northward expansion rate is not the same as lag. Thus, your conclusion "As a result, the biospheres role in future high latitude temperature amplification may be delayed." does not derive directly from the data you present.

Thank you for allowing us to clarify our wording. We have rephrased the sentence to align with our results more closely as stated in the Abstract of the manuscript.

.Line 133: From Figure S3, it looks like three or possibly four samples failed.

Thank you for pointing this out. First, we realized that the DNA reads originally included in Figure S3 were from after filtering. We have now updated the figure to include the total raw reads, which is a better indicator of DNA reliability (Rijal et al., 2021). Based on the total raw reads, only two samples failed with total reads of 2 and 5.

Line 141: You say you focus on presence/absence, but you do show quantitative results for Salix and Betula (0-5 PCR repeats) in Figure 2.

Thank you for allowing us to clarify. Fig 2 shows the number of replicates that meet our criteria for taxa presence, where a higher number of replicates corresponds to a higher likelihood of presence.

Line 142: Where are the other 51 taxa shown?

We are providing the full DNA record in the supplement, which will be published alongside the main manuscript. We have also now included a plot of species richness against sample depth in Fig. S2.

Line 178-179: Note that the revised date of first detection is close to what has been previously published (Salix ~10300 vs. 10227, Betula ~9500 vs 9680), so it does not make any changes to previous interpretation.

Yes, this is true. However, we still believe it is important to always consider improvements in age models to best correlate the timing of events between different paleo records.

Line 191-194 and Figure S2: I leave the evaluation of revised age-depth model to the geologist.

As this aspect was not commented on, we assume that both reviewers are satisfied.

Line 197: "Delay" is a more correct word than "lag".

Thank you, edited.

Line 210: Where do 1700 and 2500 come from? If your revised age of ice retreat is 11 800, and your revised date of Salix and Betula arrival are ~10 300 and ~9500, I make this 1500 and 2300.

Yes, this is correct. Thank you for pointing out this error.

Line 215-217: To be more certain about any bias caused by low DNA quality, I suggest you explore your data using the tools presented in Rijal et al. 2021 Science Advances. As you do not provide your data, I cannot evaluate the quality of them.

Thank you for the suggestion. We have now calculated the various DNA quality indices developed by Rijal et al. (2021). This has been added to the methods and results section for the Stóra Viðarvatn record, as well as in Fig. S3. The MTQ and MAQ scores are known to correlate with species richness when richness is low (n<30, Rijal et al., 2021), which is likely an artifact of the requirement that the 10 best represented barcode sequences are required to calculate these scores. As this correlation is observed in our dataset and given that our species richness is low (n<30, Fig. S2), the low MTQ and MAQ score are not likely indicative of low-quality DNA. We therefore judge the quality of our DNA on total raw reads and CT values, which remain relatively constant through time (Fig. S2).

Line 226: Do you mean TDV?

We intended to omit unnecessary abbreviations throughout the manuscript, such as lake names, in our original manuscript. We have now changed TORF, which we use as the lake’s abbreviation, to the full lake name, Torfdalsvatn.

Line 282-283: Given that the basal sediments of Nordivatnet are marine (Brown et al.2022 PNAS Nexus), even a low detection may be a strong indication of local presence.

Thank you for this point. However, to standardize the records and compare across a wide range of geographical and depositional settings, we prefer to apply the same criteria for the taxa’s presence to each lake as outlined in our Methods.

Line 289: See the definition of "lag"

Changed to “delayed” per your earlier suggestion. Thank you.

Line 298-303: I agree that the late appearance of Betula at Langfjordvatnet (10 000 cal BP) is anomalously long and a bit unexpected given that it is found at five other lakes in the region 13000-10200 cal BP (Alsos et al. 2022). However, a likely explanation is the lack of area with stable soil - B. nana requires a greater degree of soil development compared to other heath shrubs (Whittaker 1993) and Langfjordvatnet is surrounded by steep scree slopes (Otterå 2012 master thesis Univ. Bergen). At Jøkelvatnet, Salix appears in the four available samples from 10453 to 9811 whereas Betula arrives 9663. Here, the arrival of Betula is just at the drop of local glacier activity and at the temperature rise, suggesting that it arrives immediately after the climate becomes suitable (Elliott et al. 2023 Quaternary). Thus, based on N Fennoscandia where we have more data available, it does not show lags and does not support delayed shrubification (which contrasts with what we have shown for many other species including common dwarf shrubs, see Alsos et al. 2022). Would be very interesting to have similar data from Iceland, which has a large dispersal barrier.

Thank you for these further considerations. We have incorporated those related toLangfjordvannet into the manuscript accordingly. We also appreciate the point regarding Jøkelvatnet. However, as stated in our Methods section for “Published sedaDNA datasets”, we do not include Jøkelvatnet in our comparison due to the impact of glacier activity as the reviewer notes: “Finally, both Jøkelvatnet and Kuutsjärvi were impacted by glacial meltwater during the Early Holocene when woody taxa are first identified (Wittmeier et al., 2015; Bogren, 2019), and thus the inferred timing of plant colonization is probably confounded in this unstable landscape by periodic pulses of terrestrial detritus.” Due to the glacier’s presence in the lake catchment, it is not possible to discern whether delay in Betulaceae would have occurred if the glacier were not present. Therefore, we prefer to keep this record excluded from our comparisons.

Line 316-319 and 344: Based on contemporary genetic patterns, Alsos et al. analyse the relative importance of adaptation to dispersal compared to other factors.

Thank for you bringing up this important point. We have now expanded our discussion to include these analyses from Alsos et al. (2022).

Line 342+350: Original publication is Alsos et al. 2007 Science

Thank you, edited.

Line 343: Alsos et al. 2009 Salix study is the wrong citation here. Eidesen et al. 2015 Mol. Ecol. shows phylogeography of Greenland population but not Baffin - I am not aware of any contemporary genetic studies of Betula from Baffin.

Thank you for pointing this out. We will also include the Eidesen et al. (2015) citation for reference to Greenland. However, there is one data point included for southern Baffin Island in Alsos et al. (2009), so we will retain this citation here as well.

Line 351-353: See comment about Betula from Baffin above. Also, I am not sure I follow here - what do you mean by "these populations" - the Svalbard ones or Iceland? Eidesen et al. 2015 is the wrong citation for Salix - use Alsos et al. 2009. Alsos et al. 2009 suggest Iceland (and E Grenland) was colonized from north Scandinavia, although this was uncertain as no data were available from Faroe/Shetland. Svalbard was colonized from N Fennoscandia (Alsos et al. 2007).

Regarding Baffin Island sources, we refer the reviewer to our response to their previous comment. We have clarified the wording of our sentence from “these populations” to “the modern populations from these locations [Baffin Island, Greenland, and Svalbard]”. We have removed reference to Eidesen et al. (2015), as this is for Betula rather than Salix. Finally, we have added a citation for Alsos et al. (2007) here for Svalbard.

Line 354-355: AFLP suggest that Baffin and W Greenland were colonised from a refugia south of the Wisconsin Ice Sheet, see Alsos et al. 2009.

Yes, we are aware, thank you. Our reference to “mid-latitude North America” in the sentence acknowledges this refugia, but we have now added “south of the Laurentide Ice Sheet” for further clarification.

Line 363-381: See comment above; your Store Vidarvatn data do currently not demonstrate a lag between environmental suitability and climate, but using the rest of the DNA record, potentially it could. Would also be good to reflect on the distance to the source area for shrubs Late Glacial/Early Holocene compared to now.

Thank you for these suggestions. We have edited this section of the manuscript to elaborate on the need for independent climate reconstructions as well as the fact that distances to plant refugia are shorter now than during the last postglacial period.

Line 396-416: I am not an expert on tephra so I will not comment on this part.

As this aspect was not commented on, we assume that both reviewers are satisfied.

Line 459-457: Please provide results of how much data is lost at each step of filtering.

We added the read loss following each filtering step as a table in the supplemental information (Table S4).

Throughout the manuscript, you go only to species level although DNA in most cases is able to distinguish to genus level within Salicaceae and Betulaceae - which sequences did you identify?

Sequences are now provided in the supplemental for Salicaceae and Betulaceae. Based on our bioinformatic pipeline, reference library and requirement for 100% match between sequence and taxonomy, we were only able to distinguish between species level.

Figure 2: The detection of Betulaceae is very sporadic in Stóra Vidarvatn with occurrence in only seven samples and hardly ever in all 5 repeats, suggesting that if you apply a statistical model to estimate first arrival (see Alsos et al. 2022), you will have a large confidence interval. Thus, these uncertainties should be considered when estimating the delayed arrival of Betula compared to Salix. The data from Torfdalsvatn (which I assume are from Alsos et al. 2021 although not specified in the figure legend), shows detection in all samples from the first appearance and mostly in 8 of 8 repeats (shown in the original publication - you could to the same here), thus providing a more accurate estimate for the time gap between arrival of Salix and Betula.

Thank you for bringing up this important point. The detection of Betulaceae is indeed sporadic, but we believe it reflects the genuine nature of its presence/absence during the Holocene in Northeast Iceland. This is supported by Betula pollen from a nearby peat record that shows a similar history (Fig. 4, Karlsdóttir et al., 2014), which we have now elaborated on in the Results and Interpretation section. As for the timing of Betulaceae colonization at this site, the first appearance in the DNA record should be a close minimum estimate as shown with modern DNA and plant survey comparisons (e.g., Sjögren et al., 2017; Alsos et al., 2018). The true first appearance could be biased by small amounts of plants being present in the early stages of colonization and not registering the sedimentary record until enough dead plant material is transported to the depocenter of the lake. However, this is likely less than age model uncertainties and therefore not likely relevant on geologic timescales as in this study. In this sense, our age models and those published for the other records indicate this is generally on the order of several hundred years. In addition, we have now added the Alsos et al. (2021) reference for Torfdalsvatn. Unfortunately, this Torfdalsvatn study does not provide number of PCR repeats so we will keep the figure as is as it best represents the available data.

Figure 5: I suggest adding lake names to the figure. Is there a dot missing for lake 5 for Salicaceae?

Thank you for the suggestion, we have added lake names to the figure. There is a dot marked for Salicaceae for lake 5, however, not for Betulaceae as this taxon was not identified. We refer the reviewer to the Discussion Section “Postglacial sedaDNA records from the circum North Atlantic” and the lake’s original publication (Volstad et al., 2020).

Figure 6: I find it more relevant to plot colonization time versus distance to LGM sheetice margin - lake number is just an arbitrary number.

We appreciate the suggestion and have modified the figure accordingly.